EMBO
Molecular Medicine

# *CHCHD10* mutations promote loss of mitochondrial cristae junctions with impaired mitochondrial genome maintenance and inhibition of apoptosis

Emmanuelle C Genin[1], Morgane Plutino[1], Sylvie Bannwarth[1,2], Elodie Villa[3], Eugenia Cisneros-Barroso[4], Madhuparna Roy[5], Bernardo Ortega-Vila[4], Konstantina Fragaki[1,2], Françoise Lespinasse[1], Estefania Pinero-Martos[4], Gaëlle Augé[1,2], David Moore[6,7], Florence Burté[6,7], Sandra Lacas-Gervais[8], Yusuke Kageyama[5], Kie Itoh[5], Patrick Yu-Wai-Man[6,7], Hiromi Sesaki[5], Jean-Ehrland Ricci[3], Cristofol Vives-Bauza[4] & Véronique Paquis-Flucklinger[1,2,*]

## Abstract

*CHCHD10*-related diseases include mitochondrial DNA instability disorder, frontotemporal dementia-amyotrophic lateral sclerosis (FTD-ALS) clinical spectrum, late-onset spinal motor neuropathy (SMAJ), and Charcot–Marie–Tooth disease type 2 (CMT2). Here, we show that CHCHD10 resides with mitofilin, CHCHD3 and CHCHD6 within the "mitochondrial contact site and cristae organizing system" (MICOS) complex. *CHCHD10* mutations lead to MICOS complex disassembly and loss of mitochondrial cristae with a decrease in nucleoid number and nucleoid disorganization. Repair of the mitochondrial genome after oxidative stress is impaired in *CHCHD10* mutant fibroblasts and this likely explains the accumulation of deleted mtDNA molecules in patient muscle. *CHCHD10* mutant fibroblasts are not defective in the delivery of mitochondria to lysosomes suggesting that impaired mitophagy does not contribute to mtDNA instability. Interestingly, the expression of *CHCHD10* mutant alleles inhibits apoptosis by preventing cytochrome *c* release.

**Keywords** *CHCHD10*; mitochondria; mitochondrial disease; motor neuron disease; mtDNA instability

**Subject Categories** Genetics, Gene Therapy & Genetic Disease; Metabolism; Neuroscience

## Introduction

Seminal studies over the past two decades have identified a growing list of nuclear-encoded mitochondrial genes linked to mitochondrial DNA (mtDNA) instability, either due to mtDNA depletion syndrome or disorders characterized by multiple deletions (Copeland & Longley, 2014). It was predictable that nuclear genes involved in mtDNA replication and repair, or the maintenance of the intramitochondrial nucleotide pool would be implicated in human disorders characterized by mtDNA instability. However, the link between disturbed mitochondrial dynamics and human disease only became apparent with the description of the varied neurological manifestations associated with *MFN2* and *OPA1* mutations (Rouzier *et al*, 2012; Burté *et al*, 2015). More recently, we identified *CHCHD10* as a new gene responsible for mtDNA instability disorder (Bannwarth *et al*, 2014). We described a novel heterozygous *CHCHD10* mutation (c.176C>T; p.Ser59Leu) in a large French family with a phenotype including cognitive decline resembling frontotemporal dementia (FTD), motor neuron disease (MND), cerebellar ataxia, and mitochondrial myopathy with multiple mtDNA deletions. The association of FTD with MND in this family led us and others to sequence *CHCHD10* in cohorts of patients with frontotemporal dementia-amyotrophic lateral sclerosis (FTD-ALS) or with pure familial or sporadic ALS. Fascinatingly, *CHCHD10* mutations were identified in these independent cohorts, firmly establishing a pathophysiological link with FTD-ALS clinical spectrum (Chaussenot *et al*, 2014; Johnson *et al*, 2014; Müller *et al*, 2014; Chiò *et al*, 2015; Kurzwelly *et al*, 2015; Ronchi *et al*, 2015; Zhang *et al*, 2015). Furthermore, Penttilä and

1   IRCAN, UMR CNRS 7284/INSERM U1081/UNS, School of Medicine, Nice Sophia-Antipolis University, Nice Cedex 2, France
2   Department of Medical Genetics, National Centre for Mitochondrial Diseases, Nice Teaching Hospital, Nice Cedex 2, France
3   INSERM U1065, Centre Méditerranéen de Médecine Moléculaire (C3M), équipe "contrôle métabolique des morts cellulaires", Nice Sophia-Antipolis University, Nice Cedex 2, France
4   Research Health Institute of Palma (IdISPa), Research Unit, Son Espases University Hospital, Palma de Mallorca, Spain
5   Department of Cell Biology, Johns Hopkins University School of Medicine, Baltimore, MD, USA
6   Wellcome Trust Centre for Mitochondrial Research, Institute of Genetic Medicine, International Centre for Life, Newcastle University, Newcastle upon Tyne, UK
7   Newcastle Eye Centre, Royal Victoria Infirmary, Newcastle upon Tyne, UK
8   Joint Center for Applied Electron Microscopy, Nice Sophia-Antipolis University, Nice Cedex 2, France
    *Corresponding author. Tel: +33 4 93 37 77 86; Fax: +33 4 93 37 70 33; E-mail: paquis@hermes.unice.fr

colleagues identified a founder mutation in *CHCHD10* (c.197G>T; p.Gly66Val) in 17 Finnish families with late-onset spinal motor neuropathy (SMAJ) (Penttilä *et al*, 2015), and this variant is also responsible for Charcot–Marie–Tooth disease type 2 (CMT2) (Auranen *et al*, 2015).

*CHCHD10* encodes a mitochondrial protein located in the inter-membrane space and enriched at cristae junctions but the role played by this protein in both normal and disease states has not yet been established (Bannwarth *et al*, 2014). Here, we show that CHCHD10 is a component of the mitochondrial contact site and cristae organizing system (MICOS) complex (Pfanner *et al*, 2014). The expression of *CHCHD10* mutant alleles leads to MICOS complex disassembly, loss of mitochondrial cristae, and nucleoid disorganization leading to a defect in mtDNA repair after oxidative stress. Interestingly, the expression of *CHCHD10* mutant alleles inhibits apoptosis by preventing cytochrome *c* release. Our findings support previous studies suggesting that, in some ALS models, motor neuron death can occur via caspase-independent apoptotic mechanisms.

# Results

## CHCHD10 is a component of the MICOS complex that is destabilized in *CHCHD10* mutant fibroblasts

We previously found that CHCHD10 was enriched in the vicinity of mitochondrial cristae junctions as reported for mitofilin, a major component of the MICOS complex (Jans *et al*, 2013). In order to determine whether CHCHD10 is a component of MICOS, we performed blue native PAGE analysis followed by 2D Western blotting. Results in mouse brain demonstrated that CHCHD10 forms part of a multiprotein complex containing the inner membrane mitochondrial proteins mitofilin, CHCHD3, and CHCHD6 (Fig 1A).

We analyzed the effects of *CHCHD10* mutant allele on MICOS complex in fibroblasts from our original family carrying the p.Ser59Leu *CHCHD10* mutation (Bannwarth *et al*, 2014). In our first study, we analyzed fibroblasts from patient V-10 (P1). Fibroblasts from a second patient (IV-3; P2) have since become available and they showed the same pathological abnormalities, namely a multiple respiratory chain deficiency (Appendix Table S1), mitochondrial ultrastructural alterations, and fragmentation of the mitochondrial network (Appendix Fig S1). The expression of matrix-targeted photoactivatable GFP showed that mitochondrial fusion is not inhibited in fibroblasts of this novel patient (Appendix Fig S2). To confirm the absence of fusion defect in patient fibroblasts, we ectopically expressed a dominant negative mutant of the fission protein Drp1 (Drp1$_{K38A}$) (Smirnova *et al*, 1998) and found that the control and patient cells similarly elongate mitochondria (Appendix Fig S3).

Blue native PAGE followed by 2D Western blotting confirmed that, in human fibroblasts, CHCHD10 resides with mitofilin, CHCHD3, and CHCHD6 in the MICOS complex. We also observed a partial disassembly of the complex in fibroblasts of both patients compared with control cells (Fig 1B and C). The protein levels of assembled CHCHD10 and CHCHD3 were decreased in patient fibroblasts (Fig 1C). BN-PAGE analysis also revealed an impairment of OXPHOS complex IV (CIV) assembly. Consequently, OXPHOS

supercomplex formation was impaired in both patients, as evidenced by resolving the small supercomplex III+IV by lauryl maltoside solubilization (Fig 1B) and the large supercomplexes I+III$_2$+IV$_n$ (SC) by digitonin solubilization (Fig 1D). We then evaluated whether CHCHD10 could physically interact with mitofilin. We found that CHCHD10 and CHCHD3 were immunoprecipitated by a rabbit polyclonal anti-mitofilin antibody both in control and patient fibroblasts. In reverse experiment, mitofilin was immunoprecipitated by a rabbit polyclonal anti-CHCHD10 antibody in HeLa cells (Fig 2A). Last, we used a proximity ligation assay (PLA Duolink) to analyze *in situ* the proximity between CHCHD10 and mitofilin comparing control and patient fibroblasts. A similar experimental protocol was applied to study the proximity between CHCHD10 and CHCHD6. PLA experiments resulted in a positive labeling in control cells, but quantitative analysis revealed a significant decrease of dot number in patient fibroblasts (Fig 2B). Taken together, these results show that CHCHD10 is an important component of MICOS complex and that *CHCHD10* mutation leads to MICOS complex disassembly.

## Decrease in nucleoid number without mtDNA depletion in *CHCHD10* mutant fibroblasts

*CHCHD10* mutations lead to MICOS complex disassembly and loss of mitochondrial cristae. It has been suggested that there may be a threshold for the density of cristae junctions required for the maintenance of nucleoid distribution (Itoh *et al*, 2013). Based on these observations, we visualized nucleoids in patient cells carrying the p.Ser59Leu mutation to determine whether nucleoid disorganization could be responsible for the defect in mtDNA maintenance that was observed in patient skeletal muscle biopsies (Bannwarth *et al*, 2014). Immunostaining of control and patient fibroblasts with an anti-DNA antibody revealed a significant decrease of nucleoid number in patient cells compared with control cells (Fig 3A and B). Image analysis revealed no aggregation of nucleoids in patient fibroblasts despite a slight increase of nucleoid surface in cells from the first patient (Fig 3C). The phenotype was confirmed using an anti-TFAM antibody, a main protein component of the nucleoids (Appendix Fig S4). Importantly, total mtDNA content did not decrease in *CHCHD10* mutant fibroblasts (Fig 3D). Thus, the reduction of nucleoid number is not related to a reduction in the amount of mtDNA. Furthermore, it does not lead to a decrease in expression level of proteins encoded by mtDNA in patient fibroblasts (Appendix Fig S5).

## Expression of *CHCHD10* mutations in HeLa cells also leads to a decrease in nucleoid number without mtDNA depletion

To substantiate our findings, we also analyzed nucleoid characteristics in HeLa cells expressing *CHCHD10* mutant alleles. Among 94 FTD-ALS patients, we previously identified two unrelated cases who carried a novel missense mutation (c.100C>T; p.Pro34Ser) (Chaussenot *et al*, 2014), which was also reported in one Italian patient with sporadic ALS (Ronchi *et al*, 2015). To confirm the pathogenic role of this mutation, we analyzed, first, the effects of overexpression of the pathogenic allele on the morphology of the mitochondrial network. After transfection with either empty vector or the wild-type allele, MitoTracker staining revealed a filamentous network. Overexpression of mutant CHCHD10$^{P34S}$ altered

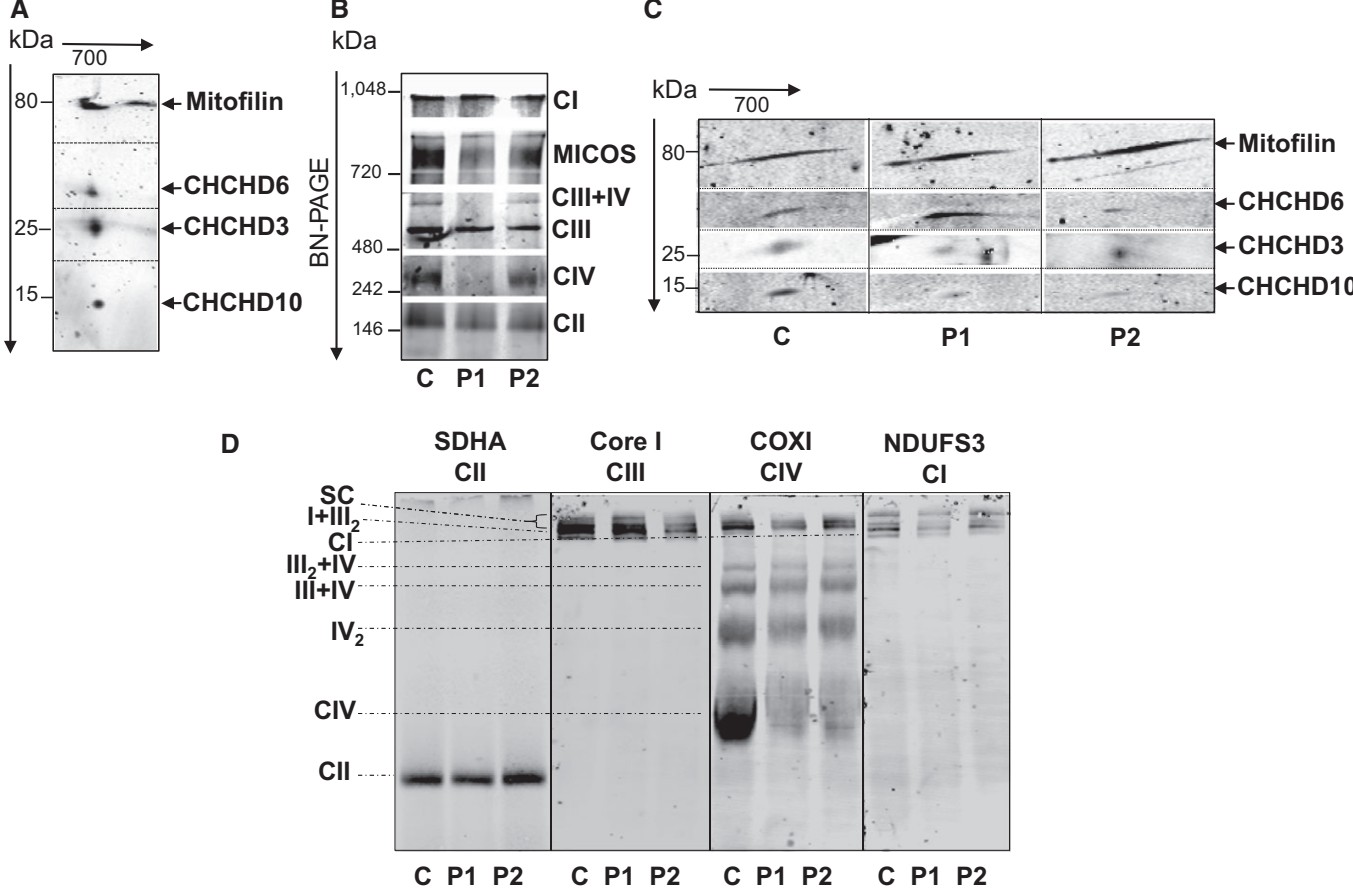

**Figure 1.  CHCHD10 resides in the same complex as mitofilin, CHCHD3 and CHCHD6.**

A   Second dimension of the blue native (BN)-PAGE showing that CHCHD10 migrates with the MICOS protein subunits mitofilin, CHCHD3 and CHCHD6 in isolated mitochondria from mouse brain.

B   BN-PAGE of the MICOS and OXPHOS complexes in control (C) and patient (P1, P2) fibroblasts. Complexes I to IV of OXPHOS were detected with the 39 kDa subunit antibody (CI), 70 kDa subunit antibody (CII), core I antibody (CIII), and cytochrome *c* oxidase subunit I antibody (CIV). MICOS complex was detected with an anti-mitofilin antibody.

C   Second dimension of the BN-PAGE showing that the steady-state levels of assembled CHCHD10 in MICOS complex are decreased in patient fibroblasts. The dividing lines correspond to gel sections visible in the raw data file.

D   Analysis of OXPHOS supercomplexes in control and patient fibroblasts. BN-PAGE from isolated mitochondria permeabilized with 6 g/g (w/v) of digitonin immunoblotted on PVDF membrane and incubated with the indicated antibodies. SC, supercomplexes I+III$_2$+IV$_n$. The dividing lines correspond to gels sections visible in the raw data file.

Source data are available online for this figure.

mitochondrial morphology in transfected cells with a significant fragmentation of the network (Appendix Fig S6). We also looked at the mitochondrial morphology by electron microscopy. Contrary to overexpression of the wild-type allele, the overexpression of the CHCHD10$^{P34S}$ mutant led to a marked defect of the mitochondrial cristae maintenance characterized by loss and disorganization of cristae morphology (Appendix Fig S7). These results are strikingly similar to those observed when we expressed the p.Ser59Leu *CHCHD10* mutation (Bannwarth *et al*, 2014). Zhang and colleagues also identified the p.Pro34Ser variant in one Canadian patient with Parkinson's disease and in two cases of Alzheimer's disease among cohorts of 153 and 141 patients, respectively (Zhang *et al*, 2015), and they suggested that it may be not pathogenic. Our data therefore confirm conclusively the deleterious effect of the p.Pro34Ser variant.

Both p.Ser59Leu and p.Pro34Ser mutations were then expressed in HeLa cells. Immunostaining with anti-DNA or anti-TFAM antibodies showed a similar pattern to that found in patient fibroblasts with a reduction of nucleoid number, but with no aggregation (Fig 4A–C; Appendix Fig S8). Furthermore, there was no reduction of mtDNA content in cells overexpressing mutant alleles compared with cells overexpressing the wild-type CHCHD10 protein (Fig 4D).

**Alteration of nucleoid organization in *CHCHD10* mutant fibroblasts**

Mitochondrial DNA molecules are thought to be closely associated with the mitochondrial inner membrane and they are included within the insoluble fraction of mitochondria (Kanki

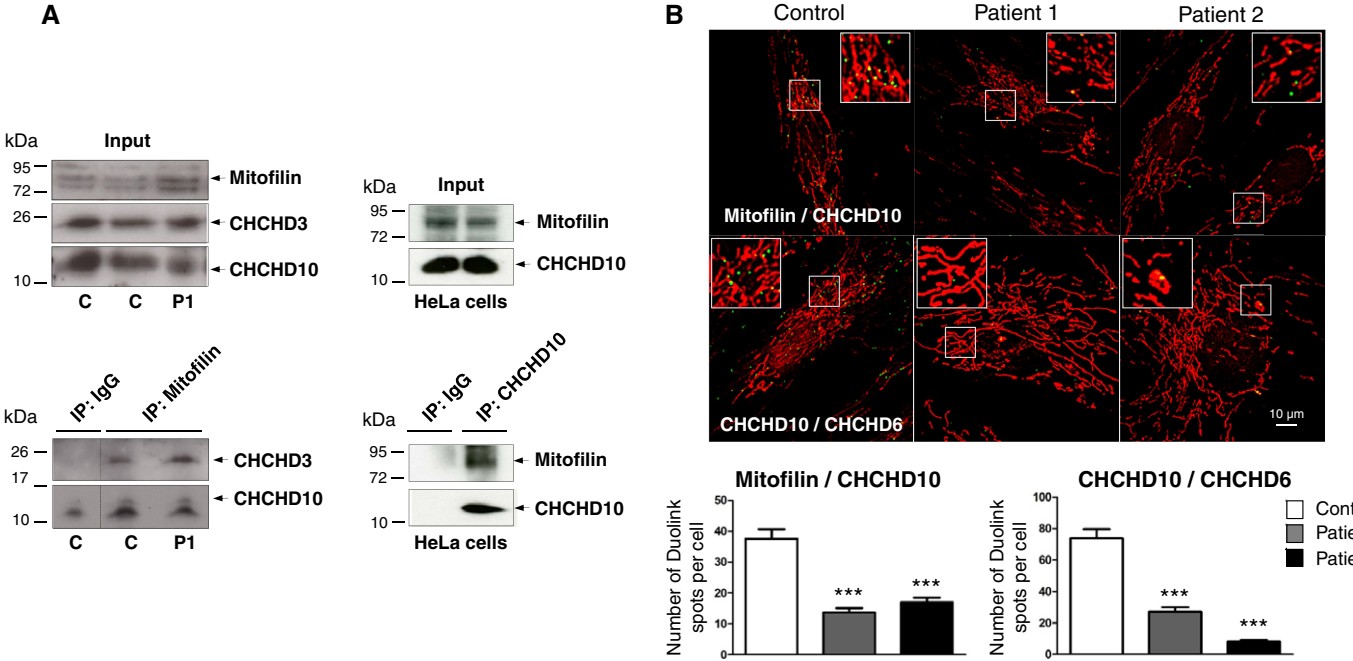

**Figure 2. Interaction of CHCHD10 with components of MICOS complex.**

A   Co-immunoprecipitation (IP) of endogenous mitofilin, CHCHD10, and CHCHD3 in control and patient fibroblasts (1 mg of total extracts was used for each IP (left lower panel) and 200 μg was used for the input (left upper panel)). The same results were found in P2 fibroblasts (not shown). Reverse co-IP experiment in HeLa cells with an antibody against CHCHD10 (right lower panel). 1 mg of total extracts was used for each IP and 200 μg was used for the input (right upper panel). The dividing lines correspond to gel sections visible in the raw data file.

B   Duolink proximity ligation assay between mitofilin and CHCHD10 (upper panels), and CHCHD10 and CHCHD6 (lower panels) in control and patient fibroblasts observed by confocal microscopy. Mitochondria were stained with MitoTracker. Duolink spots per cell were quantified for 30 randomly selected individual cells per each studied fibroblast cell line from two independent experiments. Differences between the cell lines were analyzed by Student's *t*-test (two-sided): highly significant (***P = 0.0001). Scale bar = 10 μm.

Source data are available online for this figure.

---

*et al*, 2004). In order to determine whether *CHCHD10* mutations alter the organization of nucleoids, we tested the distribution of mtDNA in Nonidet P-40 (NP-40) extraction obtained from the mitochondrial fraction of both control and patient fibroblasts. Ratio analysis of mtDNA amplified by qPCR from the supernatant and from the mitochondrial pellet showed that mtDNA was recovered from the particulate fraction in control cells, whereas mtDNA was partially released into the soluble fraction in patient cells. These results are consistent with a disturbed organization of nucleoids (Fig 5A).

**The p.Ser59Leu *CHCHD10* mutation does not influence mtDNA copy number but impairs mtDNA repair capacity under conditions of oxidative stress**

We tested the influence of the p.Ser59Leu mutation on the mtDNA copy number in patient fibroblasts following exposure to reactive oxygen species (ROS). ROS have been observed to act as a key modulator regulating mtDNA copy number in cells (Hori *et al*, 2009). We treated both control and patient cells with $H_2O_2$ and allowed 1, 2, or 4 h to recover before being harvested for DNA isolation. We compared mtDNA copy number by qPCR analysis (Sarzi *et al*, 2007). As previously reported, during the DNA recovery process, mtDNA copy number decreases over time probably due to mtDNA degradation (Hori *et al*, 2009). Monitoring of mtDNA level during the recovery time did not reveal any significant difference between control and patient fibroblasts (Fig 5B).

We then investigated whether mtDNA instability observed in patients carrying the p.Ser59Leu *CHCHD10* mutation could be secondary to a failure of mtDNA repair. Fibroblasts from control individual and patients were treated with $H_2O_2$ to introduce oxidative DNA lesions, followed by incubation to allow DNA repair. Lesions in mtDNA were assessed by gene-specific qPCR-based assay in which base lesions, abasic sites, or strand breaks interfere with the amplification of long DNA targets. This validated assay has proven particularly useful in examining mtDNA damage and repair kinetics after $H_2O_2$ treatment (Santos *et al*, 2006). Long PCR products (15.6 kb) were used and normalization was performed with shorter control amplicons (172 bp). Control fibroblasts showed a capacity to repair stress-induced DNA lesions with a 70% recovery observed 4 h after $H_2O_2$ treatment, whereas patient fibroblasts showed a severe mtDNA repair deficiency with less than 40% of recovery occurring 4 h after treatment (Fig 5C). These results suggest that *CHCHD10* mutation contributes to mtDNA instability by impairing mtDNA repair capacity after oxidative stress via nucleoid disorganization.

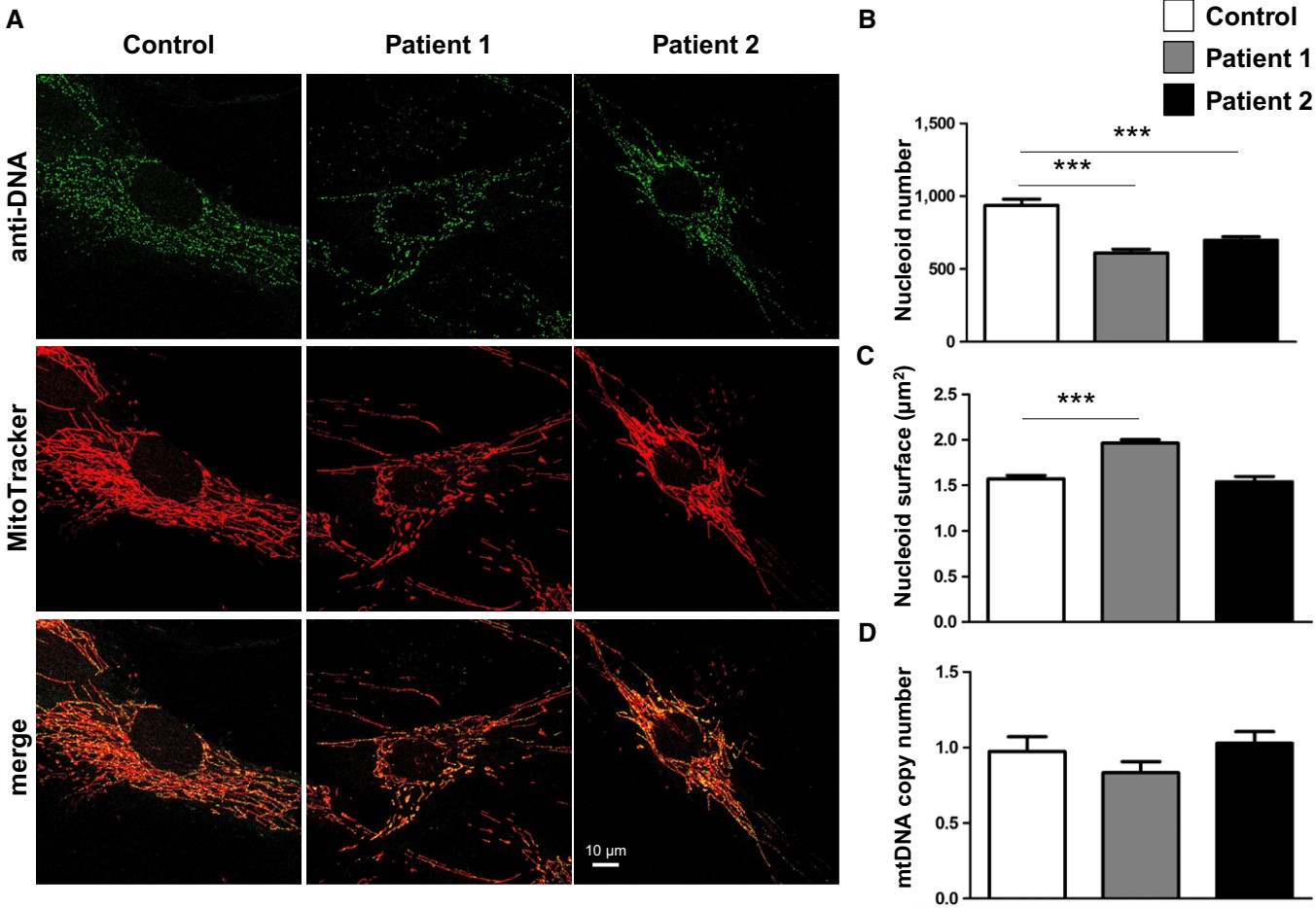

**Figure 3.   Decrease of nucleoid number without mtDNA depletion in patient fibroblasts.**

A     In control and patient fibroblasts, nucleoids were labeled with an antibody against DNA and mitochondria were stained with MitoTracker. Image analysis was performed by confocal microscopy. Scale bar: 10 μm.

B, C   The average number (B) and size (C) of nucleoids, labeled with antibodies against DNA as shown in (A), was quantified for 35 randomly selected individual cells per each studied fibroblast cell line from two independent experiments. Differences between the cell lines were analyzed by Student's $t$-test (two-sided): highly significant (***$P < 0.001$). Nucleoid number (B): patient 1 versus control ***$P = 0.0001$, patient 2 versus control ***$P = 0.001$. Nucleoid size (C): patient 1 versus control ***$P = 0.0001$.

D     Mitochondrial DNA content in control and patient fibroblasts was quantified by real-time PCR. Data were expressed as ratio between mtDNA and nuclear DNA concentration. Results represent the mean of relative PCR $\pm$ SD of three independent experiments. Statistical analyses were performed by Student's $t$-test (two-sided).

## CHCHD10 mutant fibroblasts display no impact on general autophagy rate and they are not defective in the delivery of mitochondria to lysosomes

The accumulation of mtDNA deletions in skeletal muscle could also be secondary to a failure to clear mitochondria with damaged DNA (Chen *et al*, 2010). We compared the expression of the autophagy marker LC3B between fibroblasts from patients and controls, in basal conditions and after treatment with chloroquine (Fig 6A). Quantification of LC3B-II/β-tubulin ratio revealed no significant difference between patients and controls (Fig 6B). As expected, chloroquine did not inhibit the expression of LC3B-II because, as a lysosomal inhibitor, it would favor its accumulation. Quantification of the ratio between LC3B-II levels within samples at baseline and after treatment with chloroquine also revealed no

difference between patient and control cells (Fig 6C). These results suggest that, under basal conditions, the p.Ser59Leu mutation does not inhibit general autophagy rate in patient fibroblasts. To gain further insight into the effects of the p.Ser59Leu mutation on mitophagy, we monitored the delivery of mitochondria to lysosomes using a biosensor for this process, mt-Keima (Katayama *et al*, 2011). mt-Keima is a variant of RFP that is fused to a presequence to the mitochondrial matrix. This cellular biomarker changes its fluorescent profile in response to pH and it is resistant to degradation within lysosomes. We infected control and patient fibroblasts using lentiviruses carrying mt-Keima and analyzed the intracellular distribution of mt-Keima by measuring the fluorescence intensity in mitochondria and in lysosomes (Fig 6D). Control and patient fibroblasts showed not significant difference in $FL_{lyso}/FL_{mito}$ ratios suggesting that the p.Ser59Leu mutation does

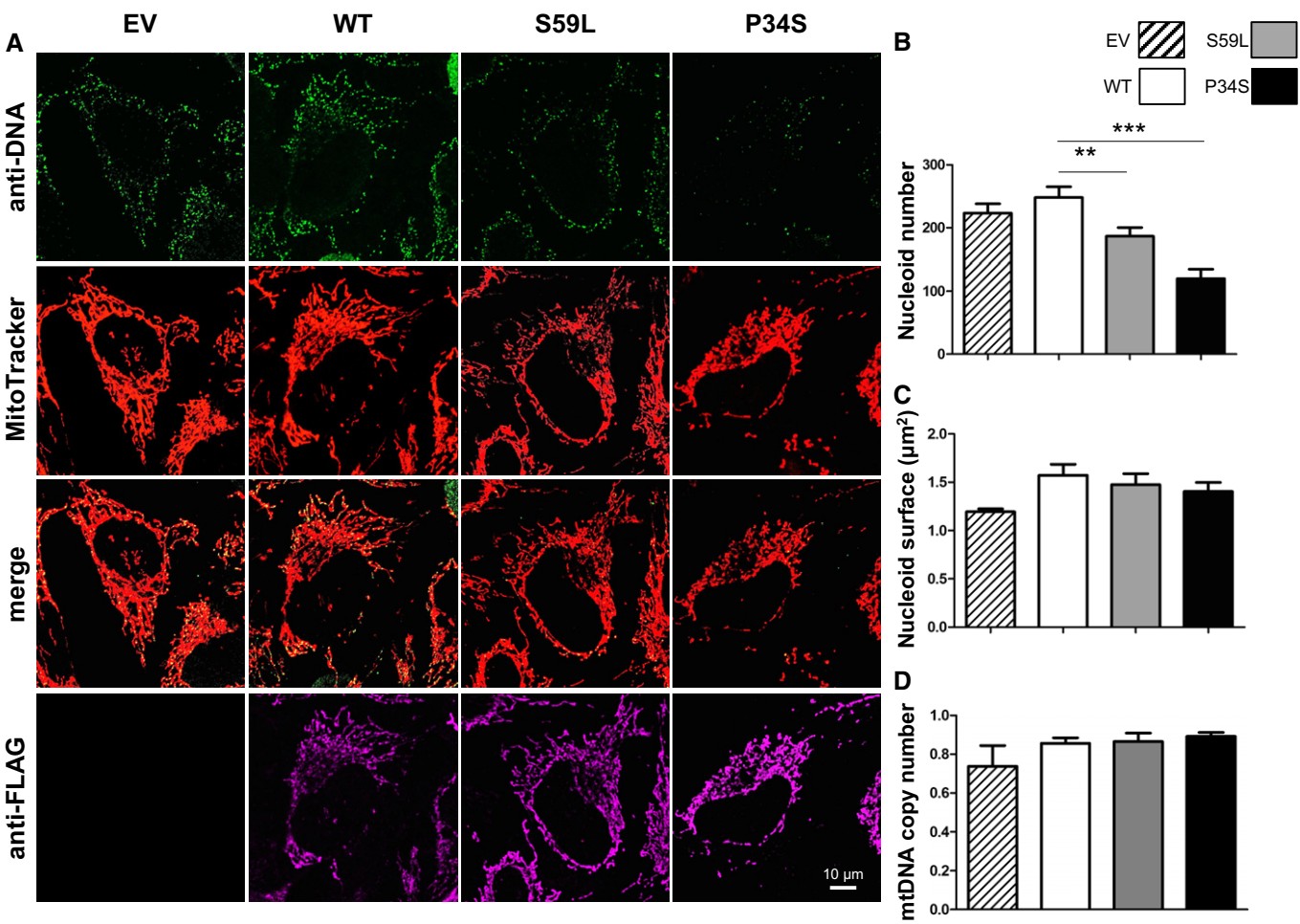

**Figure 4.  Decrease of nucleoid number without mtDNA depletion in HeLa cells overexpressing the *CHCHD10^{S59L}* and *CHCHD10^{P34S}* alleles.**

A    Transfections were performed with empty vector (EV) or vectors encoding wild-type CHCHD10-FLAG (WT), CHCHD10-FLAG (S59L), or CHCHD10-FLAG (P34S) mutants. Mitochondria were stained with MitoTracker. Cells overexpressing wild-type and mutant CHCHD10 were labeled with FLAG antibodies. Nucleoids were visualized with an antiserum against DNA. Image analysis was performed by confocal microscopy. Scale bar: 10 μm.

B, C    The average number (B) and size (C) of nucleoids, labeled with antibodies against DNA as shown in (A), was quantified for 35 randomly selected individual cells per each studied cell line from two independent experiments. Differences between the cell lines were analyzed by Student's *t*-test (two-sided): very significant (**$P$ = 0.0280) or highly significant (***$P$ = 0.0001).

D    Mitochondrial DNA content was quantified by real-time PCR. Data were expressed as ratio between mtDNA and nDNA concentration. Results represent the mean of relative PCR ± SD of three independent experiments. Statistical analyses were performed by Student's *t*-test (two-sided).

not impact on the delivery of mitochondria to lysosomes in fibroblasts (Fig 6E).

**Inhibition of apoptosis and delayed cytochrome *c* release in cells expressing *CHCHD10* mutant alleles**

Finally, we investigated whether *CHCHD10* mutations could lead to disease state secondary to disturbed apoptotic mechanisms. Annexin V/DAPI staining was performed in primary fibroblasts isolated from healthy individuals (controls) or from the two patients (V-10 and IV-3) carrying the p.Ser59Leu mutation. Mutant fibroblasts were significantly less sensitive to staurosporine-induced apoptosis compared with controls (Fig 7A). This reduction in annexin V/DAPI staining was associated with a reduction in PARP cleavage and of caspase activation. The latter was visualized with a reduction

of the cleaved form of caspase-3 as well as by a decrease in cellular DEVDase activity (Fig 7B and C). These results were confirmed in HeLa cells expressing the WT, CHCHD10^{S59L} or CHCHD10^{P34S} mutant forms (Fig 8A–C). We further observed that mutant fibroblasts did not present a difference in mitochondrial membrane potential (ΔΨm) under basal conditions. However, following staurosporine treatment, although the ΔΨm dropped dramatically in control fibroblasts as was expected, mutant cells were less affected (Fig 7D and E). Furthermore, release of cytochrome *c* was delayed following staurosporine in patient fibroblasts compared to control cells (Appendix Fig S9). These results are in line with the reduction of cell death observed in the mutant fibroblasts.

Then, using HeLa cells expressing the WT, CHCHD10^{S59L}, or CHCHD10^{P34S} mutant forms, we showed that this reduction in apoptosis was associated with a reduction in the permeabilization of the

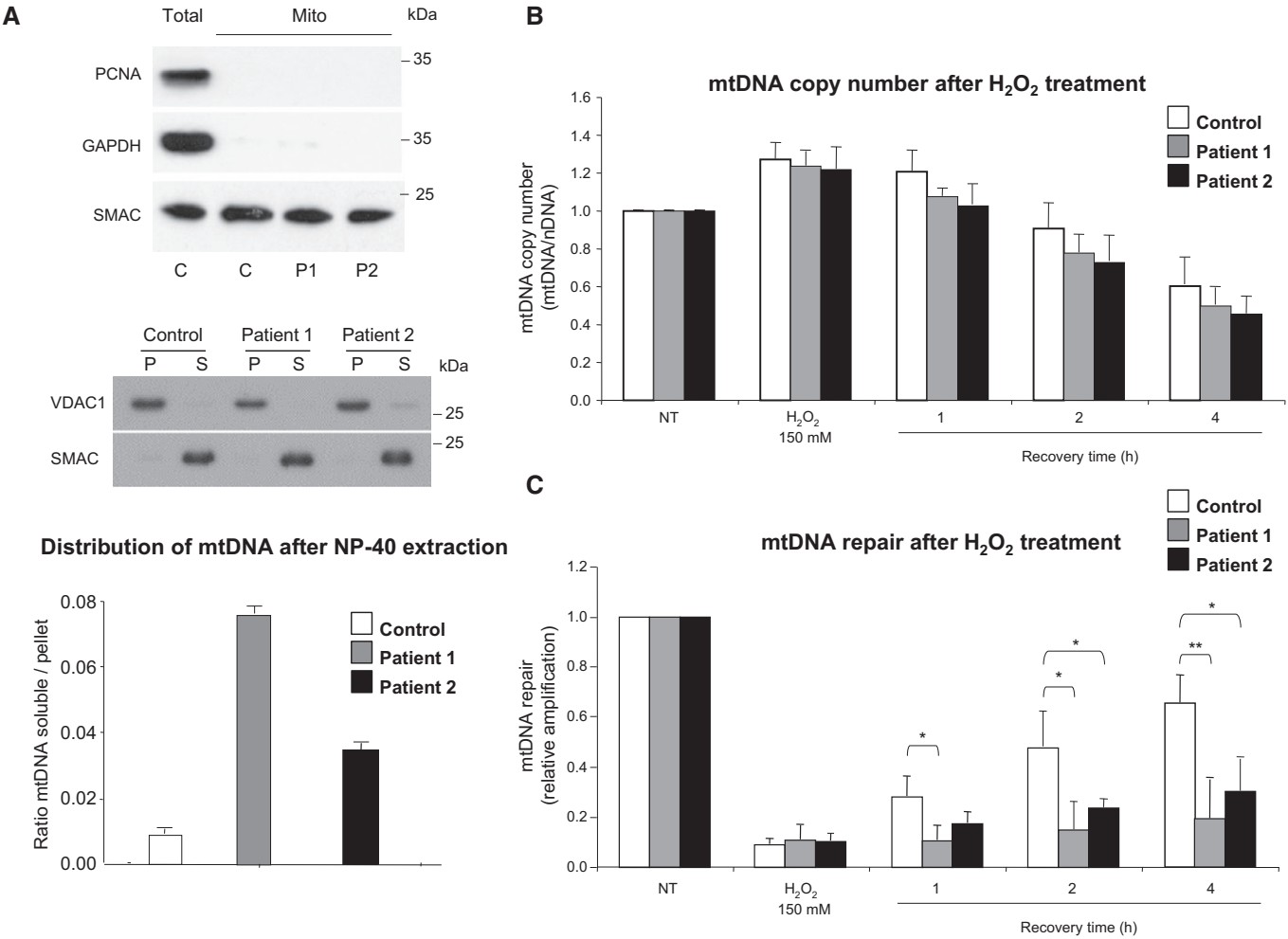

**Figure 5. Nucleoid disorganization in patient fibroblasts leading to a defect in mtDNA repair under conditions of oxidative stress.**

A   Total extracts from control (C) fibroblasts and intact isolated mitochondria from control and patient fibroblasts (P1, P2) were analyzed by immunoblotting using antibodies against PCNA (nuclear protein), GAPDH (cytosolic protein), or SMAC (mitochondrial intermembrane space protein) (upper panel). Mitochondria from patient and control fibroblasts were incubated with NP-40 and separated into pellets (P) and supernatants (S). The fractions of each extraction were subjected to Western blot analysis. VDAC and SMAC were used to identify behaviors of well-defined mitochondrial proteins that are integral membrane and soluble proteins, respectively (middle panel). Ratio of mtDNA amplified from supernatant/mtDNA amplified from pellet, by qPCR, was quantified in control and patient fibroblasts (lower panel).

B   Determination of the mtDNA copy number after $H_2O_2$ treatment. mtDNA/nDNA values in control, patient 1, and patient 2 fibroblasts. mtDNA: mitochondrial DNA, nDNA: nuclear DNA.

C   mtDNA repair after $H_2O_2$ treatment. A long-range PCR was used to evaluate the oxidative damage, induced by $H_2O_2$ treatment, in mtDNA. The relative PCR amplification of a 15.6 kb mtDNA fragment was normalized to mtDNA copy number that was evaluated by PCR amplification of a 172-bp mtDNA fragment. mtDNA repair activity in control, patient 1, and patient 2 fibroblasts.

Data information: In (B, C), cells were exposed to 150 μM $H_2O_2$ for 30 min and either harvested immediately or allowed to recover in conditioned medium for the indicated times. Untreated control cultures were incubated in serum-free medium alone. Results represent the mean of relative PCR amplification ± SD of three independent experiments in which three PCRs per point were performed. Values were normalized to untreated cells and differences were analyzed by Student's *t*-test (two-sided): significant (*: 0.05 > P > 0.01), very significant (**: 0.01 > P > 0.001). Patient 1 versus control: *P = 0.015 (recovery 1 h), *P = 0.022 (recovery 2 h), **P = 0.003 (recovery 4 h). Patient 2 versus control: *P = 0.0230 (recovery 2 h), *P = 0.041 (recovery 4 h). NT: cells not treated with $H_2O_2$.

Source data are available online for this figure.

outer membrane of the mitochondria (MOMP), with SMAC degradation being used as a readout of MOMP (Fig 8D) (Tait *et al*, 2010). The expression of $CHCHD10^{S59L}$ or $CHCHD10^{P34S}$ mutant alleles also delayed cytochrome *c* release compared with HeLa cells expressing wild-type CHCHD10 (Appendix Fig S10).

Taken together, our experimental data indicate that expression of both the p.Ser59Leu and p.Pro34Ser *CHCHD10* mutations is able to decrease the sensitivity of the cells toward apoptotic stimuli.

## Discussion

Recent studies in yeast have identified the mitochondrial inner membrane organizing system (MINOS) complex, which is located within the inner mitochondrial membrane (IMM) (Von der Malsburg *et al*, 2011; Friedman *et al*, 2015). This complex was recently renamed to MICOS and a nomenclature for its subunits was proposed using the 3 letters Mic followed by the apparent molecular

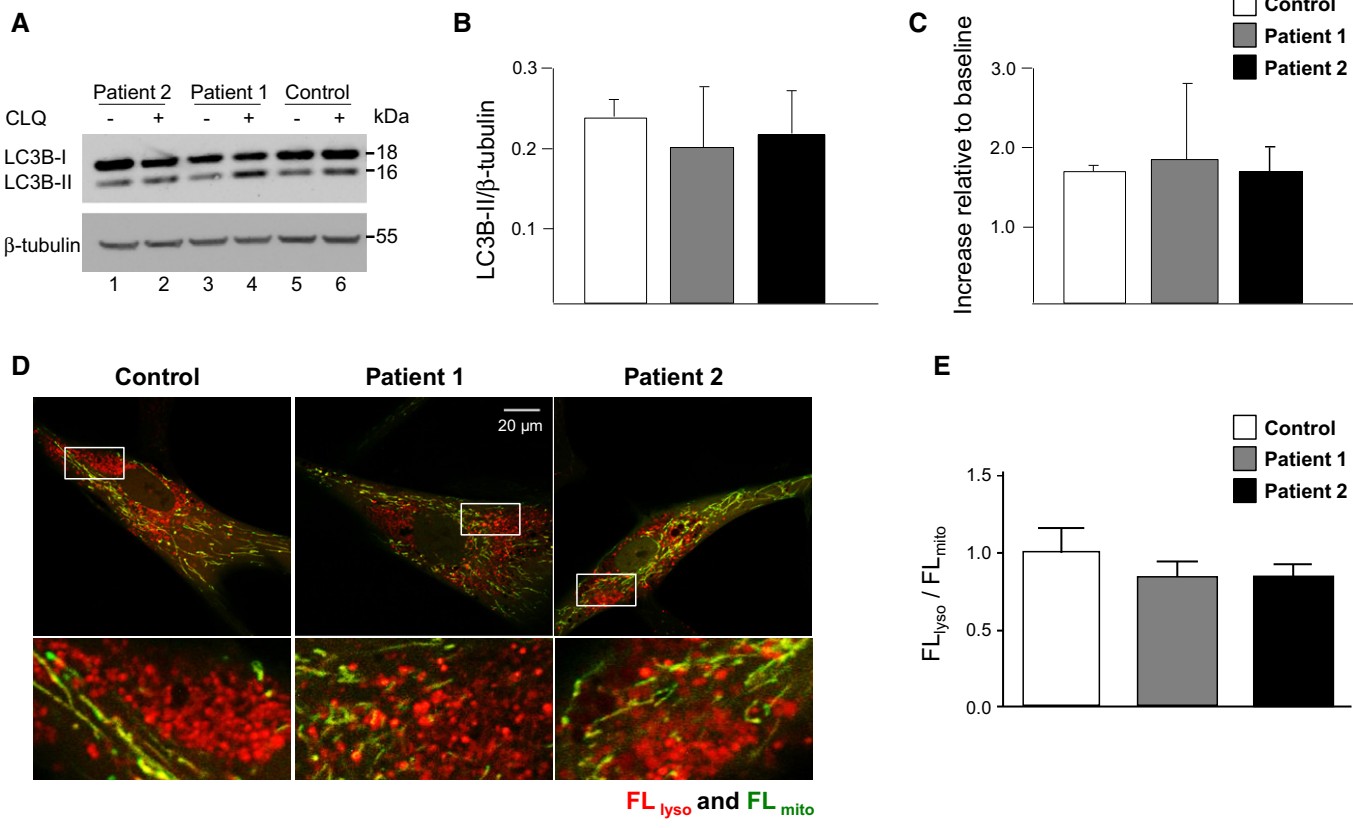

**Figure 6.   No autophagy inhibition and no mitophagy defect in patient fibroblasts.**

A   Immunoblot of LC3B-I/-II expression in fibroblasts of patient 2 (1, 2), patient 1 (3, 4), and control individual (5, 6) in the absence (−) or presence (+) of chloroquine (CLQ).

B   Quantitative analysis of LC3B-II/β-tubulin ratio from immunoblots (*n* = 3).

C   Quantitative analysis of LC3B-II + CLQ / LC3B-II − CLQ ratio from immunoblots (*n* = 3).

D   Confocal images of fibroblasts derived from a control, patient 1, and patient 2. Boxed regions are enlarged. Scale bar: 20 μm.

E   Bar graph showing ratios of fluorescent intensity of mitochondria delivered to lysosomes relative to that of mitochondria present in the cytosol (FL$_{lyso}$/FL$_{mito}$).

Data information: Statistical analyses were performed on mean ± SEM using Student's *t*-test (two-sided).

Source data are available online for this figure.

weight of the protein (Pfanner *et al*, 2014). Mitofilin, newly called Mic60, is a central component of yeast MICOS complex and is associated with at least 5 different other proteins, namely Mic27, Mic26, Mic19, Mic12, and Mic10 (Harner *et al*, 2011; Hoppins *et al*, 2011; Von der Malsburg *et al*, 2011; Alkhaja *et al*, 2012; Pfanner *et al*, 2014). The core components of MICOS have been highly conserved during evolution. In mammals, 5 subunits have been demonstrated so far to constitute this complex: MIC60/Mitofilin, MIC27/APOOL, MIC25/CHCHD6, MIC19/CHCHD3, and MIC10/MINOS1. Depletion of mitofilin in human cells or deletion in yeast leads to abnormal cristae structures with a massive loss of cristae junctions (John *et al*, 2005; Rabl *et al*, 2009). Destabilization of MICOS also correlates with concomitant loss of cristae junctions, which indicates that the integrity of MICOS is required for the formation and/or maintenance of these structures (Zerbes *et al*, 2012; Friedman *et al*, 2015). We had previously shown that *CHCHD10* encodes an inter-membrane space mitochondrial protein that is enriched at cristae junctions and that expression of *CHCHD10* mutant alleles led to abnormal cristae structures with loss of cristae junctions

(Bannwarth *et al*, 2014). Our current study has confirmed these findings by showing that CHCHD10/MIC14 resides with mitofilin/MIC60, CHCHD3/MIC19, and CHCHD6/MIC25 in the MICOS complex. The expression of *CHCHD10* mutant alleles results in partial disassembly of the mitofilin/CHCHD3/CHCHD6/CHCHD10 complex, which is the triggering factor that underlies the dramatic alterations in mitochondrial cristae morphology observed in patient cells. Detachment of cristae from the inner boundary membrane likely leads to a separation of the mitochondrial respiratory chain complexes from the protein import machinery and thus from the supply of substitute proteins that is required for the replacement of damaged subunits (Zerbes *et al*, 2012). This probably explains at least in part the bioenergetic deficit associated with the expression of *CHCHD10* mutant alleles. Furthermore, loss of mitochondrial cristae junctions may have additional negative effects on the assembly of respiratory chain complexes and the stability of mitochondrial supercomplexes, which are mainly located within cristae membranes. The impairment of complex IV assembly and of super-complex formation observed in patient fibroblasts supports our

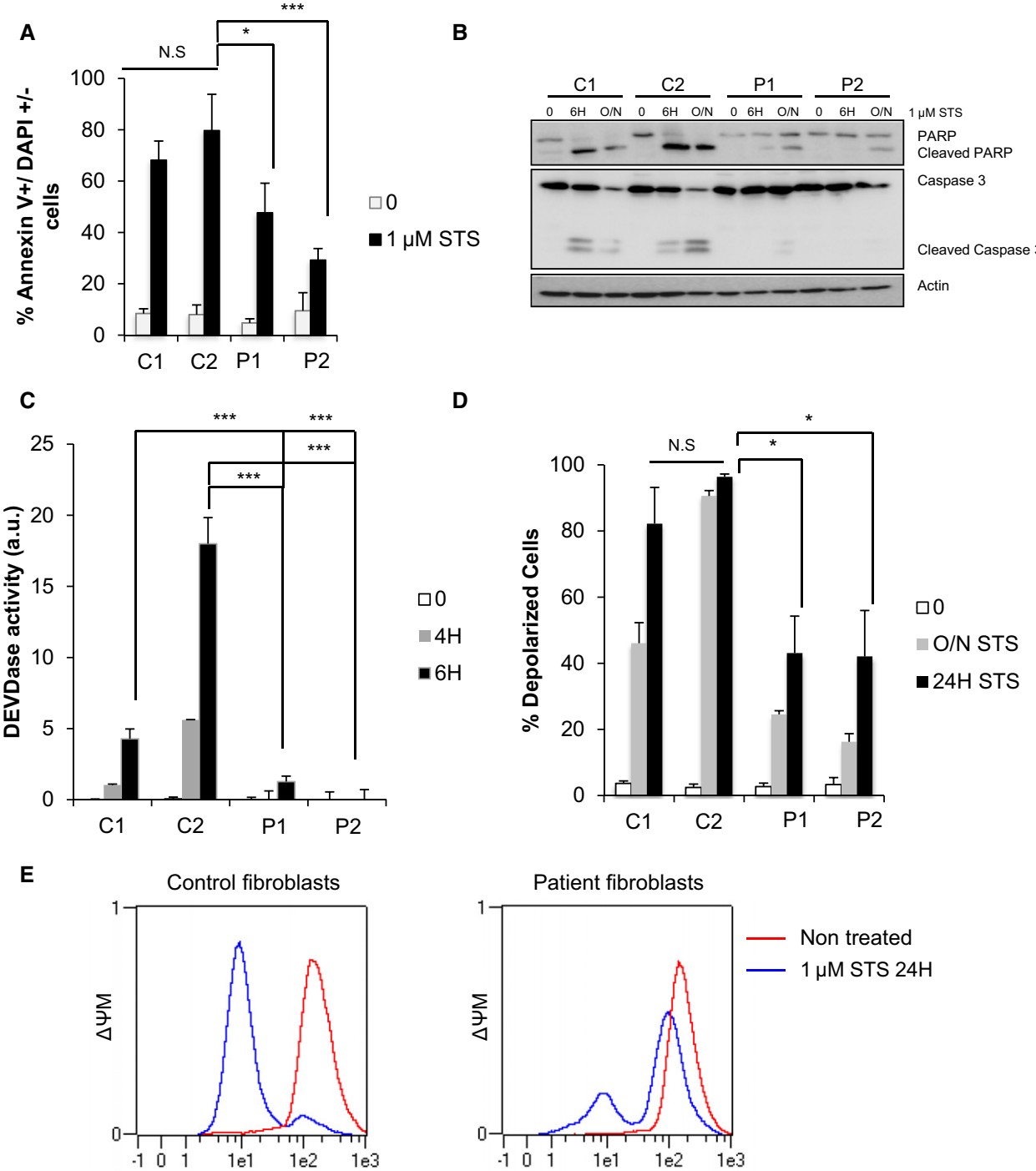

**Figure 7.  *CHCHD10* mutant fibroblasts are less sensitive to apoptotic cell death compared to control fibroblasts.**

A    Control (C1, C2) and patient (P1, P2) fibroblasts were treated with 1 μM of staurosporine (STS) for 16 h. Cell death was determined by flow cytometry using annexin V/DAPI staining. Three independent experiments were performed per condition with two points analyzed per experiment. Differences between the control and patient fibroblasts were analyzed by Student's *t*-test (two-sided): significant (*$P$ = 0.0116) or highly significant (***$P$ = 0.0001).

B, C    Fibroblasts were treated with 1 μM STS for 4, 6, or 16 h (O/N), and caspase activation was determined either by Western blot (B) assessing the cleavage of PARP and the caspase-3 (actin was used as a loading control) or by DEVDase activity measurement (C). Two (B) or three (C) independent experiments were performed per condition. Differences between the control and patient fibroblasts were analyzed by Student's *t*-test (two-sided): extremely significant (***$P$ = 0.0001).

D, E    Mitochondrial membrane potential (ΔΨM) was measured by flow cytometry in control and patient fibroblasts after treatment with 1 μM STS for 24 h (E) and quantified from two independent experiments (D). Differences between the control and patient fibroblasts were analyzed by Student's *t*-test (two-sided): significant (*: 0.05 > $P$ > 0.01). Patient 1 (P1) versus control (C2): *$P$ = 0.0414 (24 h). Patient 2 (P2) versus control (C2): *$P$ = 0.0461 (24 h).

Source data are available online for this figure.

    

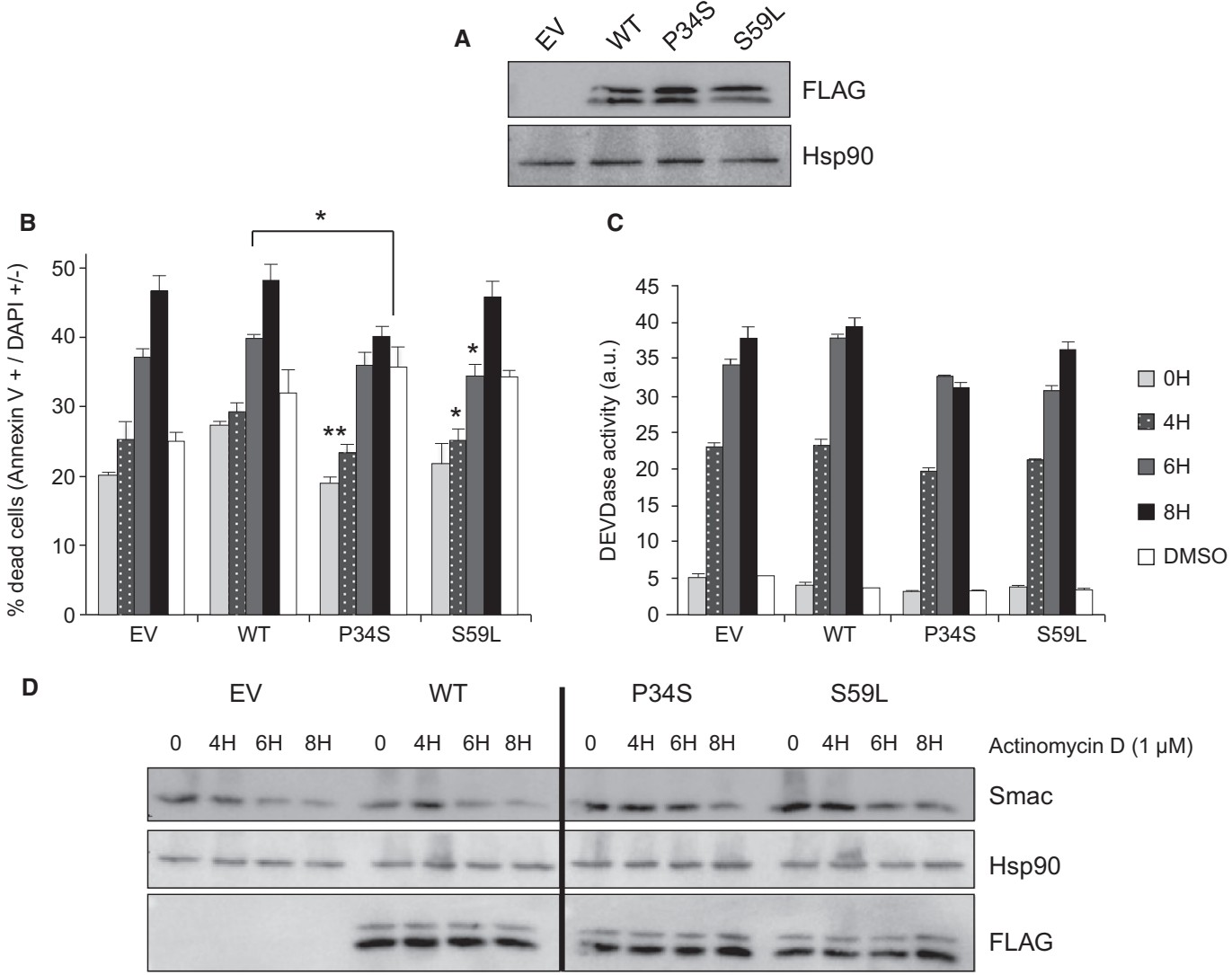

**Figure 8.  Overexpression of the *CHCHD10* mutant alleles in HeLa cells leads to inhibition of apoptotic cell death.**

Transfections were performed with empty vector (EV) or vectors encoding either wild-type CHCHD10-FLAG (WT) or mutant CHCHD10-FLAG (P34S and S59L).

A     Western blot on HeLa cell extracts using antibodies against FLAG or Hsp90.
B–D   HeLa cells transiently expressing the WT or mutated forms of CHCHD10-FLAG (P34S or S59L) were treated with 1 μM actinomycin D (ActD) for 4, 6, or 8 h with measurement of annexin V/DAPI staining (B), DEVDase activity (C) from three independent experiments. Differences between the mutated and non-mutated alleles were analyzed by Student's *t*-test (two-sided): significant (\*: 0.05 > *P* > 0.01) or very significant (\*\*: 0.01 > *P* > 0.001). P34S versus WT: \*\**P* = 0.0055 (4 h), \**P* = 0.0103 (8 h). S59L versus WT: \**P* = 0.0304 (4 h), \**P* = 0.0129 (8 h). Mitochondrial outer membrane permeabilization was determined by Western blot by assessing the degradation of SMAC (D). Hsp90 was used as a loading control.

Source data are available online for this figure.

hypothesis and this would account, at least partly, for the multiple respiratory chain deficiency observed (Bannwarth *et al*, 2014).

The accumulation of multiple mtDNA deletions is a characteristic pathological hallmark of disorders of mitochondrial maintenance (Copeland & Longley, 2014). Multiple mtDNA deletions were found in muscle biopsies from patients carrying the p.Ser59Leu mutation, and our data show that this is not due to disturbed mitochondrial dynamics as mitochondrial fusion is not inhibited in patient fibroblasts. Since a minimum density of cristae junctions is thought to be required for the maintenance of nucleoid distribution (Itoh *et al*, 2013), we analyzed the effects of *CHCHD10* mutations on nucleoid organization. Mitochondrial nucleoids carry 2 to 10 mtDNA molecules packaged with proteins, including mitochondrial transcriptional factor A (TFAM), and they appear to be a platform for transcription, translation, replication, and repair of mtDNA (Alam *et al*, 2003). In human fibroblasts, nucleoids are formed by clusters of small uniform structures consisting of a single copy of mtDNA and with an average of 1,184 ± 59 nucleoids per cell with DNA antibodies and 964 ± 50 nucleoids per cell with TFAM antibodies by confocal microscopy (Kageyama *et al*, 2014). We found similar results in control fibroblasts, but there was a significant decrease in the number of nucleoids in patient cells. Importantly,

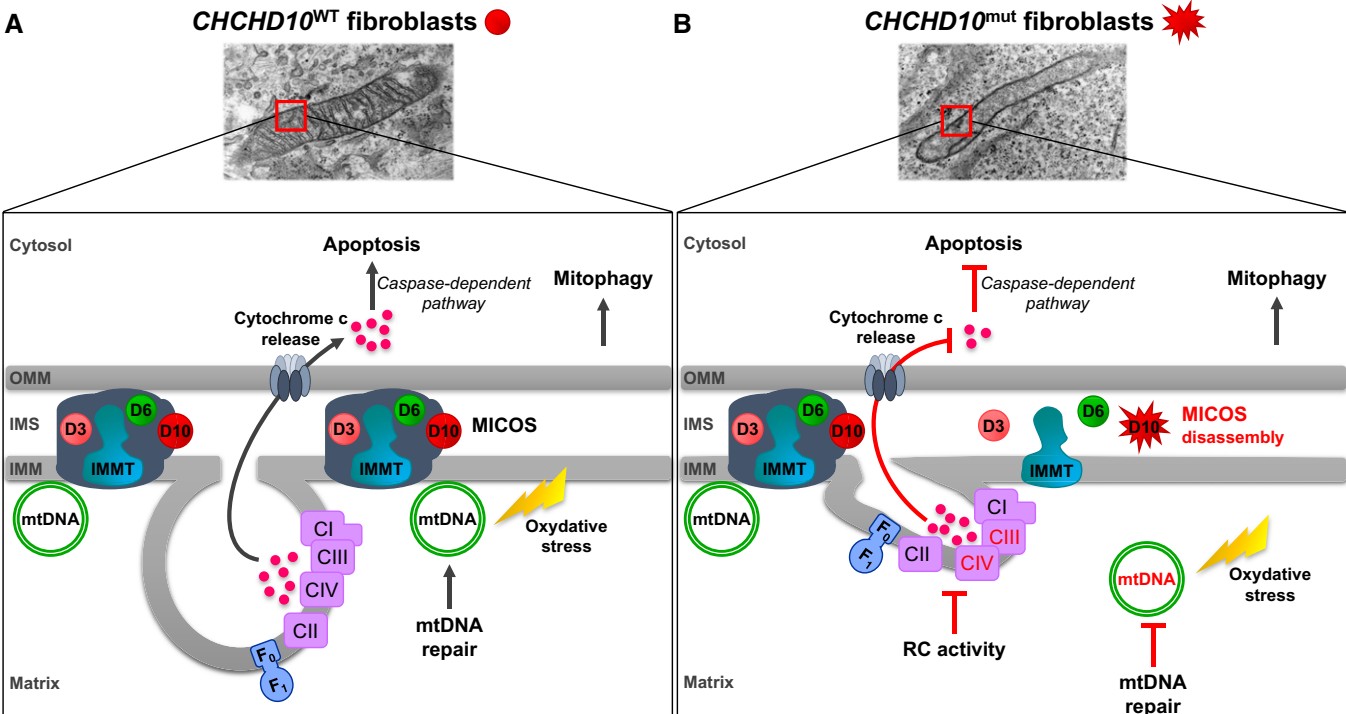

**Figure 9.** **Model for the differential actions of wild-type (WT) and mutant CHCHD10 proteins in mitochondria.**

A  Model of WT CHCHD10 function. CHCHD10 (D10) resides with mitofilin (IMMT), CHCHD3 (D3), and CHCHD6 (D6) in the mitochondrial inner membrane organizing system (MICOS) complex.

B  Model of deleterious effects of mutant CHCHD10. *CHCHD10* mutations lead to MICOS complex disassembly and loss of mitochondrial cristae with impairment of complex IV (CIV) assembly and CIII + CIV supercomplex formation leading to a respiratory chain (RC) deficiency. Nucleoids are disorganized in *CHCHD10* mutant cells resulting in defect in mtDNA repair after oxidative stress. *CHCHD10* mutations have no impact on mitophagy. However, the expression of *CHCHD10* mutant alleles prevents cytochrome *c* release and thus inhibits apoptosis via the caspase-dependent pathway. OMM, outer mitochondrial membrane; IMS, intermembrane mitochondrial space; IMM, inner mitochondrial membrane.

total mtDNA content was not decreased both in patient fibroblasts and in HeLa cells overexpressing *CHCHD10* mutant alleles. We observed no enlarged nucleoids that could explain the reduction of nucleoid number, a phenomenon that has been reported previously in the context of Drp-1-mediated mitochondrial fission defects (Ban-Ishihara *et al*, 2013). In *CHCHD10* mutant fibroblasts, the decrease in nucleoid number is likely secondary to their physical disorganization in relation to the inner mitochondrial membrane. Indeed, nucleoids are mainly membrane associated and most mtDNA is recovered in the particulate insoluble fraction when extracted from normal mitochondria with NP-40 (Alam *et al*, 2003). However, when isolated from *CHCHD10* mutant fibroblasts, mtDNA was partially released into the soluble fraction indicating that the organization of the mitochondrial nucleoids has been altered. This disorganization is likely to have a serious impact on mtDNA repair mechanisms, especially under conditions of oxidative stress, and it is an attractive mechanism that would explain the accumulation of multiple mtDNA deletions in skeletal muscle of patients harboring pathogenic *CHCHD10* mutations. The CHCHD10 protein resides in the MICOS complex with mitofilin, which has also been identified in nucleoid isolation experiments (Wang & Bogenhagen, 2006; Jans *et al*, 2013). One could therefore speculate that *CHCHD10* mutations are responsible for nucleoid disorganization via MICOS complex destabilization and the subsequent loss of cristae junctions.

Mitochondrial autophagy is a protective mechanism that allows for the removal of mitochondria harboring high levels of mtDNA damage (Meyer & Bess, 2012). It is therefore conceivable that deficient mitophagy is another mechanism contributing to the mtDNA instability associated with pathogenic *CHCHD10* mutations. We found no impact on general autophagy rate in patient fibroblasts, and importantly, there was no impaired delivery of mitochondria to lysosomes. The presence of abnormal protein aggregates or inclusions containing specific misfolded proteins is a common feature of most neurodegenerative disorders. Many studies in cellular and animal models of motor neuron disease indicate an enhanced autophagy activity (Nassif *et al*, 2010). However, mutations in two genes associated with ALS, charged multivesicular body protein-2B (*CHMP2B*) and the lipid phosphatase Fig4 *(FIG4)* lead to severe impairment of the autophagic pathway, strengthening the hypothesis that autophagy defect may contribute to ALS (Lee *et al*, 2007; Ferguson *et al*, 2009). Furthermore, impairment of the ubiquitin–proteasome system, but not the autophagy–lysosome system, in motor neuron replicates ALS in mice (Tashiro *et al*, 2012). These contrasting observations on the role of autophagy in ALS are perhaps not surprising as ultimately, the same neurodegenerative disorder can be the consequence of diverse pathological pathways, depending largely on the predominant genetic defect driving cellular dysfunction and neuronal loss. Although further experiments in mouse and other cellular

models are necessary, our results suggest that the mtDNA instability observed in skeletal muscle of patients carrying the *CHCHD10* p.Ser59Leu mutation is not associated with impaired mitophagy.

Since mitochondrial cristae sequester the bulk of cytochrome *c* molecules, we also explored the impact of *CHCHD10* mutations on apoptosis. The expression of wild-type CHCHD10 does not protect HeLa cells from apoptosis induced by actinomycin D. Surprisingly, expression of CHCHD10$^{S59L}$ or CHCHD10$^{P34S}$ mutant forms leads to a significant reduction in cell death. In keeping with this observation, patient fibroblasts carrying the p.Ser59Leu mutation were significantly less sensitive to staurosporine-induced apoptosis compared with control cells. While excessive apoptosis has been associated with neurodegenerative disorders, as one would expect, it is becoming abundantly clear that apoptosis is not the only cellular mechanism that regulates cell death and drives neuronal loss (Agostini *et al*, 2011; Pirooznia *et al*, 2014). Re and colleagues reported that human adult astrocytes, which have been derived from motor cortex and spinal cord of sporadic ALS patients, have a deleterious influence on human embryonic stem-cell-derived motor neurons by triggering a form of regulated necrosis, termed necroptosis, in the absence of caspase activation (Re *et al*, 2014). The wobbler mouse is one of the best-characterized model of spontaneous motor neuron degeneration and it mimics several of the key features seen in ALS patients (Moser *et al*, 2013). Several studies suggest that motor neuron death occurring in these mice is not related to caspase-dependent apoptotic mechanisms (Popper *et al*, 1997; Ballinger *et al*, 1999). Lastly, "dying-back" mechanism, where distal axon degeneration occurs early during the disease, before neuronal degeneration, and the onset of overt clinical symptoms, has been reported in mouse models and patients with ALS (Dadon-Nachum *et al*, 2011). Nevertheless, the observation that the disassembly of MICOS complex linked to *CHCHD10* mutations should lead to protection against apoptosis via cristae alteration is surprising. During development, about 50% of all neurons undergo programmed cell death (PCD), a process that is critical for the establishment and maturation of a definitive pattern of neuronal connections (Yeo & Gautier, 2004). Apoptosis is also responsible for focal elimination of cells within differentiated neuronal populations and PCD appears to be critical for cell number control, quality control, and phenotypic selection. *CHCHD10* is the first gene responsible for FTD-ALS clinical spectrum and SMAJ that encodes for a primary mitochondrial protein. Although further work is needed in this area, it is tempting to speculate that *CHCHD10* mutations could negatively impact upon PCD resulting in an impaired ability to eliminate dysfunctional neurons, which in turn perpetuate a negative spiral of cell degeneration via caspase-independent mechanisms.

In conclusion, this is the first report establishing a causative link between mutations in a gene encoding for a component of the MICOS complex and human disease. Disassembly of the MICOS complex secondary to *CHCHD10* mutations leads to mitochondrial dysfunction including loss of mitochondrial cristae, respiratory chain deficiency, nucleoid disorganization with impaired mtDNA repair, and inhibition of apoptosis (Fig 9). The pathological manifestations associated with *CHCHD10* mutations provide a unique opportunity to gain further insight into the complex cellular pathways linking mitochondrial dysfunction with the broader area of neurodegeneration.

## Materials and Methods

### Blue native electrophoresis (BN-PAGE)

Mitochondrial membranes were isolated from $2.5 \times 10^6$ cells or from 200 µg of pure mitochondria as described previously (Nijtmans *et al*, 2002). Cells were solubilized with 3% digitonin (wt/vol) (Sigma-Aldrich) and 0.4% (wt/vol) lauryl maltoside (Sigma-Aldrich) or 1% digitonin (wt/vol). Pure mitochondria were solubilized with two rounds of digitonin (8 g/g and 4 g/g of protein). Ten microliters of samples were electrophoresed on a 5–13% gradient polyacrylamide gel as described previously (Nijtmans *et al*, 2002). Transfer of proteins onto a PVDF membrane (Bio-Rad Laboratories) was carried out overnight at 30 V at 4°C. For second-dimension gel electrophoresis, a lane excised from the first dimension native gel was first treated for 30 min with denaturing buffer containing 15 mM β-mercaptoethanol and 1% SDS and then washed in 1% SDS for 1 h. The gel strip was electrophoresed on a tricine–SDS–polyacrylamide gel as described previously (Ballinger *et al*, 1999).

For supercomplex analysis, enriched mitochondrial fraction was solubilized with 6 g/g digitonin. 30 µg proteins were loaded to a 4–13% Bis–Tris native gel. Electrophoresis was run at 4 mA during 23 h. Proteins were transferred at 63 mA for 24 h to a PVDF membrane.

### *In situ* proximity ligation assay (PLA)

Fibroblasts were seeded in 16-well Lab-Tek chamber slides (Nunc). For mitochondrial staining and after an attachment period of 48 h, cells were incubated in a 100 nM solution of MitoTracker red for 15 min. Then, the samples were washed twice with PBS and fixed with PFA 4% for 20 min at 37°C, washed five times with PBS, and permeabilized with 2% Triton X-100 for 10 min. After five PBS washes, coverslips were saturated with 5% BSA for 45 min at RT. The following antibodies were used in PLA assay: rabbit anti-CHCHD10 (Sigma), mouse anti-mitofilin (Abcam) or rabbit anti-mitofilin (Proteintech), and goat anti-CHCHD6 (Santa Cruz). All antibodies were diluted (1/200) with PBS–BSA 5%. Fibroblasts were incubated in the presence of convenient couple of primary antibodies for 1 h at RT. After two PBS washes, incubation with appropriate PLA probe (PLA probe anti-rabbit MINUS, PLA probe anti-mouse PLUS or PLA probe anti-goat PLUS), hybridization, ligation, and amplification were done using the DuoLink *In Situ* Detection Reagents Green (Olink Biosciences) following manufacturer's instructions. Finally, the samples were mounted on glass slides using Prolong Gold Antifade Reagent (Molecular Probes) and analyzed using a DeltaVision Imaging System (GE Healthcare Life Sciences). Data are represented as mean ± SEM. Statistical analyses were performed by Student's unpaired *t*-test using GraphPad Prism 5 (GraphPad Software). Negative control experiments (with one antibody omitted) were performed in parallel and checked to result in the absence of PLA signal.

### Preparation of mitochondrial NP-40-soluble and insoluble fractions

Mitochondria were isolated using QProteome mitochondria isolation kit (Qiagen) as described by the manufacturer. All solutions contained 1× concentration of protease inhibitor mix, CompleteMini™ (Roche). Mitochondria (5 mg/ml) were solubilized in TES

buffer (0.25 M sucrose, 10 mM Tris–HCl, pH 7.0, and 1 mM EDTA) containing 0.5% NP-40. After incubation for 30 min at 4°C with shaking, the mitochondria were centrifuged for 30 min at 10,000 $g$ at 4°C and separated into supernatant (S) and pellet (P). The pellet was resuspended in the same solubilizing buffer and was centrifuged 30 min at 10,000 $g$ at 4°C for a second wash.

### mtDNA quantification by qPCR in total mitochondria, P, and S fractions

Briefly, 10 μl of samples was mixed with 10 μl of 2× digestion buffer (100 mM Tris–HCl, pH 8.0, 2 mM EDTA, 2% SDS, and 2 mg/ml proteinase K) and incubated 30 min at 55°C. After addition of 180 μl of distilled water, the mixtures were further incubated for 10 min at 100°C to inactivate proteinase K. Samples were diluted 10-fold with distilled water. Relative mtDNA quantification of mtDNA fragment (nucleotides 1,195–1,305) was performed by qPCR. PCR mixture (20 μl) contained 4 μl of DNA samples, 1× LightCycler 480 probes master mix (Roche), 6 μM of 12S TaqMan probe (5′-6FAM-AAACCCCGATCAACC-3′MGBNFQ), and 0.3 nM of each primer (5′-TAGAGGAGCCTGTTCTGTAATCGA-3′ (forward) and 5′-TGCGCTTACTTTGTAGCCTTCAT-3′ (reverse)). PCR amplification, performed in a Light Cycler LC480 apparatus, consisted of a single denaturation–enzyme activation step for 10 min at 95°C, followed by 45 amplification cycles of 15 s at 95°C and 40 s at 60°C. A single acquisition was done at the end of each annealing step, and data were analyzed using LightCycler software version 1.5.0.39 (Roche).

### Autophagy analysis

Autophagy was assessed by Western blot analysis of the autophago-some marker LC3B-II (1/500, Cell Signaling), which correlates with the number of autophagosomes. Anti-β-tubulin (1/2,000, Abcam) antibodies were used for normalization. Autophagic flux, that defines the entire autophagy dynamic process, was assessed by the ratio of LC3B-II between samples in the presence and absence of chloro-quine, an autophagy inhibitor. Cells were exposed to chloroquine (10 μM) for 2 h prior to analysis with Western blot. Densitometry was performed using ImageJ software (National Institute of Health).

### Mitophagy analysis

Fibroblast cells were infected with lentiviruses expressing mt-Keima and observed with a dual excitation imaging system using Zeiss LSM 510 with a 60× objective lens. Images were taken at 3 days postinfection (Katayama *et al*, 2011; Kageyama *et al*, 2014). Excitation 458 nm and emission > 650 nm were used to detect mt-Keima in mitochondria in the cytosol ($FL_{mito}$, green). Excitation 561 nm and emission > 650 nm were used to detect mitochondria in the lysosomes ($FL_{lyso}$, red). mt-Keima signals in mitochondria and those in lysosomes were measured in whole cells. Background fluorescence was calculated in three independent positions in the cell, averaged and subtracted from the whole cell fluorescence using the NIH ImageJ software. The ratio of fluorescent intensity $FL_{lyso}$/$FL_{mito}$ was quantified. Values are mean ± SEM ($n$ = 3 experiments). Approximately 10 cells were analyzed in each experiment. Two-tailed $t$-test was performed. $P$-values were not significant for both patient cells compared to control cells.

### The paper explained

#### Problem
Amyotrophic lateral sclerosis (ALS) is a devastating disease affecting motor neurons and leading to progressive failure of the neuromuscular system and death from respiratory failure. Among the factors involved in ALS pathogenesis, mitochondrial dysfunction has always been recognized as a potential major player. However, whether mitochondria have a causative role in ALS has been always debated. Recently, we provided genetic basis to support the conclusion that mitochondrial dysfunction may have a causative effect in motor neuron degeneration. We reported a large family with a mitochondrial myopathy associated with motor neuron disease and cognitive decline resembling frontotemporal dementia (FTD). We identified a missense mutation (p.Ser59Leu) in the *CHCHD10* gene encoding a mitochondrial protein whose function was unknown. The association of FTD with motor neuron disease in this family led us and others to sequence *CHCHD10* in cohorts of patients with frontotemporal dementia-amyotrophic lateral sclerosis (FTD-ALS) or with pure familial or sporadic ALS. Of note, *CHCHD10* mutations were identified in these independent cohorts, firmly establishing a pathophysiological link with FTD-ALS clinical spectrum.

#### Results
Here, we show that CHCHD10 belongs to the MICOS complex, which is involved in mitochondrial cristae maintenance. Disassembly of the MICOS complex in patient fibroblasts carrying the *CHCHD10*$^{S59L}$ mutant allele leads to loss of mitochondrial cristae with disorganization of nucleoids. Repair of the mitochondrial genome after oxidative stress is impaired in patient cells and this likely explains the accumulation of deleted mtDNA molecules in patient muscle. Interestingly, the expression of *CHCHD10* mutant alleles inhibits apoptosis by preventing cytochrome *c* release.

#### Impact
Our results open new opportunities to explore the pathogenesis of motor neuron disease by showing that mitochondrial dysfunction and MICOS disassembly may underlie some of these phenotypes. They represent a fundamental step to determine which interventions, aimed at restoring correct specific mitochondrial functions, may present a therapeutic option.

Expanded view for this article is available online.

## Acknowledgements

We acknowledge Pasteur-IRCAN Cellular and Molecular Imaging platform (PICMI). This work was made possible by grants to VP-F from the Fondation pour la Recherche Médicale (FRM) and the Association Française contre les Myopathies (AFM), to HS from National Institutes of Health (GM089853) and to J-ER from la Fondation ARC pour la Recherche sur le Cancer. PYWM is supported by a Clinician Scientist Fellowship Award (G1002570) from the Medical Research Council (UK) and also receives funding from Fight for Sight (UK) and the UK National Institute of Health Research (NIHR) as part of the Rare Diseases Translational Research Collaboration. CV-B is supported by the Spanish Instituto de Salud Carlos III (CP11/00046). MP is granted by an FRM fellowship (DEA20130727390).

## Author contributions

ECG, MP, SB, KF, and FL designed and performed experiments, analyzed data, and contributed to the main text; GA performed experiments; EV performed and analyzed apoptosis experiments; J-ER designed and analyzed apoptosis experiments and contributed to the main text; EC-B, BO-V and EP-M performed and analyzed MICOS experiments; CV-B performed and analyzed

MICOS experiments and contributed to the main text; MR, YK and KI performed and analyzed mitophagy and fusion experiments; HS analyzed mitophagy experiments and contributed to the main text; DM and FB performed and analyzed autophagy experiments; PY-W-M analyzed mitophagy experiments and contributed to the main text; SL-G performed electron microscopy experiments; VP-F supervised the project and wrote the manuscript.

## Conflict of interest

The authors declare that they have no conflict of interest.

## For more information

http://www.ircan.org

http://www.ncbi.nlm.nih.gov/books/NBK304142

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
