## [Review Process File · EMBO Molecular Medicine]

CHCHD10 mutations promote loss of mitochondrial cristae junctions with impaired mitochondrial genome maintenance and inhibition of apoptosis

Emmanuelle C. Genin, Morgane Plutino, Sylvie Bannwarth, Elodie Villa, Eugenia Cisneros-Barroso, Madhuparna Roy, Bernardo Ortega-Vila, Konstantina Fragaki, Françoise Lespinasse, Estefania Pinero-Martos, Gaëlle Augé, David Moore, Florence Burté, Sandra Lacas-Gervais, Yusuke Kageyama, Kie Itoh, Patrick Yu-Wai-Man, Hiromi Sesaki, Jean-Ehrland Ricci, Cristofol Vives-Bauza, Véronique Paquis-Flucklinger

Corresponding author: Véronique Paquis-Flucklinger, IRCAN Institute, UMR7284/U1081/ University of Nice Sophia-Antipolis, Faculté de Médecine

Review timeline:

Submission date:	08 June 2015
Editorial Decision:	25 June 2015
Revision received:	02 October 2015
Editorial Decision:	20 September 2015
Revision received:	29 October 2015
Editorial Decision:	04 November 2015
Revision received:	09 November 2015
Accepted:	16 November 2015

Transaction Report:

Editor: Roberto Buccione

1st Editorial Decision

25 June 2015

Thank you again for the submission of your Review article manuscript to EMBO Molecular Medicine. We have now heard back from the three Reviewers whom we asked to evaluate your manuscript.

As you will see the two out three Reviewers are quite positive but do raise an issue of great concern. Although I will not dwell into much detail, I would like to highlight the main points.

Reviewers 1 and 3 share the most important item of concern, that one of the most interesting findings, namely that CDCHD10 is part of the MICOS complex, is far from being conclusively shown. You will see that the Reviewers offer various suggestions in this respect. In aggregate, the two Reviewers also list other important items that should not prove too difficult to address fully.

Reviewer 2 is quite negative and is generally dissatisfied with the overall quality of the data in terms of both presentation and interpretation. Although his/her tone is a bit too cursory perhaps, I would strongly recommend carefully dealing with all the points mentioned, which in part are also reflected in the other Reviewers' comments.

In conclusion, while publication of the paper cannot be considered at this stage, we would be pleased to receive a substantially revised submission, with the understanding that the Reviewers' concerns must be fully addressed with additional experimental data where appropriate and that acceptance of the manuscript will entail a second round of review.

Please note that it is EMBO Molecular Medicine policy to allow a single round of revision only and that, therefore, acceptance or rejection of the manuscript will depend on the completeness of your responses included in the next, final version of the manuscript.

As you know, EMBO Molecular Medicine has a "scooping protection" policy, whereby similar findings that are published by others during review or revision are not a criterion for rejection. However, I do ask you to get in touch with us after three months if you have not completed your revision, to update us on the status. Please also contact us as soon as possible if similar work is published elsewhere.

Please note that EMBO Molecular Medicine now requires a complete author checklist (<http://embomolmed.embopress.org/authorguide#editorial3>) to be submitted with all revised manuscripts. Provision of the author checklist is mandatory at revision stage; The checklist is designed to enhance and standardize reporting of key information in research papers and to support reanalysis and repetition of experiments by the community. The list covers key information for figure panels and captions and focuses on statistics, the reporting of reagents, animal models and human subject-derived data, as well as guidance to optimise data accessibility.

I look forward to seeing a revised form of your manuscript as soon as possible.

***** Reviewer's comments *****

Referee #1 (Remarks):

The manuscript by Genin et al., presents CHCHD10 as potential new MICOS subunit. The MICOS complex is important for the formation of cristae membranes and contact sites. The authors present a characterization of fibroblasts of patients having a point mutation within CHCHD10 and human HeLa cells overexpressing CHCHD10 mutant forms. They observed abnormal cristae formation and reduced apoptosis. The authors found that impaired repair of mtDNA after oxidative stress lead to a decreased number of mitochondrial nucleoids in CHCHD10 mutants. Overall the study involves an impressive set of state of the art techniques and the experimental quality is sound. The presented findings are potentially very interesting for the readership of EMBO Molecular Medicine. My major concern is that the binding of CHCHD10 to the MICOS complex is not convincingly shown as outlined below. This point has to be addressed before the manuscript is suitable publication.

1. In Figure 1E the association of CHCHD10 to Mic60 is shown by co-immunoprecipitation. To exclude unspecific co-elution of CHCHD10 with Mic60 IP the authors have to include a control like preimmunserum. Furthermore, it should be given in the figure legend how many percentage is loaded for the input and elution fractions to allow assessment of the efficiency of the pulldown.
2. To substantiate the association of CHCHD10 to Mic60 the authors should perform reverse pulldown experiments, e.g. using Flag-tagged CHCHD10 of HeLa cells for co-IPs.
3. The authors present a small section of a 2D PAGE to reveal co-migration of MICOS components and CHCHD10 (Figure 1A, C). To judge whether the co-migration is specific the whole second dimension has to be shown for each protein. Furthermore, the size of the markers of the first dimension should be indicated. Is it possible to shift the Mic60 signal on BN/2D in Flag-CHCHD10 using anti-Flag antibodies? Such or similar experiments would demonstrate the presence of Mic60

in the same complex as CHCHD10.

4. In Figure 1C the migration pattern of Mic60 remains unaltered in patients mitochondria. Does this mean that CHCHD10 is not important for structural integrity of MICOS? Can the authors speculate about the role of CHCHD10 within the MICOS complex?

5. In Figure 1B the complete 1st dimension should be shown for each blot.

6. The labeling of Figure 1D is not entirely clear. Which bands correspond to the labeled respiratory chain complexes?

7. The authors should show steady state protein levels of mitochondrial-encoded proteins and respiratory chain components by SDS-PAGE. This analysis will reveal whether the reduced copy number of mitochondrial nucleoids affects levels of mitochondrial-encoded proteins.

8. Does CHCHD10 show any sequence similarity to known MICOS components? Since MICOS subunits are conserved from yeast to human it would be interesting to see whether CHCHD10 is conserved as well.

Referee #2 (Comments on Novelty/Model System):

A number of the data presented and not acceptable and not of a standard that is commensurate with publication in EMBO Mol Med.

Although the findings are derived from patient lines and have points of interest, the results are not of sufficient medical relevance to be considered appropriate for the journal. I would not recommend acceptance.

Referee #2 (Remarks):

The authors report a mechanism for the action of CHCHD10 and the defects caused in samples from patients carrying mutations in the CHCHD10 gene.

- Fig 1C in which CHCHD10 is described as reduced in the P1 and P2 samples is not of the best quality. Each sample is made up of 4 separate sections, which for P1 and 2 overlap the CHCHD6 signal partially obscuring it. Because the panels for each antibody are split into separate sample panels, it is not possible to tell if all were exposed to the detection system for the same length of time or if the difference in signal is a result of different exposure times rather than reflecting the protein amounts. This is particularly important when claiming that the levels of CHCHD10 are down whilst those of CHCHD6 are up. The current data is not suitable to make the conclusions presented.

The blot for the mouse seemed to be able to detect all 4 antibody signals at the same time. A similar panel with all control and patient samples detected together would be needed to make this data convincing.

- The results for Patient 1 and 2 differ. This is not altogether surprising but they must be consistent if the message is that they both differ in a similar way to controls. Fig 2 depicts quantification indicating that there is a significant difference between both patients and control for nucleoid number but only a significant difference in size of nucleoids for Patient 1.

- Fig 4 gives tabulated values for the mtDNA in the soluble vs pellet fraction. This is meaningless, without as a minimum, a western to show that the fractionation is consistent between samples. It does separate the nuclear from the cytoplasmic component. This will not discriminate between cytosolic and mitochondrial compartments. It may be more of an indication of fractured mitochondria rather than disturbed nucleoids.

- Levels of LC3 are shown graphically and not the westerns. It is necessary to see the primary data to evaluate the validity of the graphs.

- Controls have a level of natural variation as can be seen in fig 6D where the difference in O/N STS is high. The trend following treatments is identical between patients and controls, only the absolute % of depolarized cells varies.

-

Minor comments

- In the introduction there is a rather awkward sentence that is not readily understandable" If the involvement of nuclear genes responsible for either mtDNA replication/repair or nucleotide pool maintenance was predictable, that of genes falling in other categories such as MFN2, which is involved in mitochondrial fusion, was rather unexpected."
- Fig 1 B has a horizontal arrow for kDa. This is not necessary or accurate as this represents a 1D BN PAGE expt.
- Quality of the CHCHD10 panel in Fig1E - pre clear is poor, especially compared to the adjacent panel of the IP. The indication of data not shown for P2 should be in the main text, not the legend.
- Fig 2 A - this may be a well known phenomenon but if the antibody is against DNA why does the nucleus not light up ?
- FLAG should be given in in capital letters.
- I don't really understand the axis label for the mtDNA content. "Data were expressed as ratio between mtDNA and nDNA concentration." What is regarded as a control normal value ? Do they mean concentration or amount ? In Fig2 the control is set as 1, but in Fig 3 none of the samples are given a value of 1, so what are they compared to ?

Referee #3 (Comments on Novelty/Model System):

Some of the experiments may require complementary analysis to consolidate the conclusions. Details are found in the remarks to the authors.

Referee #3 (Remarks):

This article represents a significant advance in the knowledge of CHCHD10 function and in the molecular mechanisms that connect this factor with human pathology. I think it is suitable for publication in EMBO Molecular Medicine if the considerations and comments that follow are adequately answered.

Following are my comments, doubts and suggestions for the consideration by the authors:

General considerations:

I.- I assume that the fibroblast and HeLa cell results are normally from glucose medium culture. The fibroblast model grown in glucose could be not completely appropriate to analyze some phenotypes associated with the mutations in CHCHD10, that seem to be cell-type specific. For example the deletions and complex V alterations are seen in patient muscle but not in fibroblasts. By the same token, the respiratory defect is not evidenced in fibroblasts grown in glucose and the differences in DEVDase activity after apoptosis induction, very clear in fibroblasts, are not detected in HeLa cells. Since the respiratory defect, although relatively mild, could promote a metabolic shift to a more glycolytic profile in the mutant cells, it would be interesting to see some of the analysis performed in the galactose growth medium (for example the membrane potential analysis). In this respect, do the average cell size or cell morphology change when comparing control and patient fibroblasts? And between the glucose and galactose medium?

II.- One of the conclusions of this work is that mitochondrial fusion is not affected by the mutations in CHCHD10. This is somehow unexpected in the view of the role proposed for this protein and the fact, for example, that the KO for a CHCHD10 partner, CHCHD3 (also reduced in the mutants) does present fusion alterations (Darshi et al, JBC, 2011). Since diffusion of components of the inner membrane as well as those of the nucleoids is slow (Busch et al, Phil. Trans. R. Soc. B 369: 20130442. <http://dx.doi.org/10.1098/rstb.2013.0442>, 2014), may be longer times are required to detect differences in the fusion process. In this respect, P2 seems to have at longer times a reduction in fusion (suppl. figure 2). Please comment on this. And, how do you explain the mt fragmentation pattern in the mutants, by an increase in fission? Some more work could be done in future works to clarify this point.

III. Another important conclusion in the manuscript is that CHCHD10 is a component of the MICOS complex. In a recent article, Guarani et al (eLife 2015;4:e06265. DOI:

10.7554/eLife.06265) using interaction proteomics, systematically screened for proteins that associate with cristae junctions and although they found most known and some new interactions, failed to detect CHCHD10. Although the results presented are consistent with the proposal, this could be reinforced (see also specific points). For example, the co-immunoprecipitation experiment (Fig 1E) could be done also with the anti-CHCHD10 antibody to see if mitofilin or other MICOS components are pulled-down.

Specific points:

1.- Fig 1 A. It is interesting to see tissue data in addition to cell culture, but It would be interesting to know why have they chosen mouse brain for the 2D and what is the size of the MICOS complex band containing mitofilin and CHCHD10.

2.- Fig 1B. In the case of P2 sample, I have the impression that if normalization is made by the CII level, no big decrease in MICOS) would be found. Is that correct?

3.- Fig 1C. The proportion of CHCHD3 vs D6 changes as well as the co-localization of bands. Thus, co-migration seems total in C but only partial in P1 (more clear) and P2. How are these changes in the proportion of the different components compatible with the MICOS complex keeping the same size?

4.- Fig 1 D. In the analysis of supercomplexes there is a low resolution (separation) of high MW bands (maybe this is related to the electrophoresis conditions, that are very different to the original ones proposed by Schögger). The more clear effect in complex levels is the strong reduction in free CIV (also evident for patient 1 in fig 1B). I am surprised that this reduction has no clear correlation with the complex activities shown in suppl. Table 1. In particular, I find unexpected that P1 (Bannwarth et al. 2014) shows a higher activity in COX than P2 having a more severe reduction in CIV. It would be interesting to see also here data on coupled/uncoupled respiration and in galactose medium.

5.- Figure 2-3, (analysis of nucleoids):

Why anti-DNA antibody detect less dots in mutant cells if total mtDNA is unchanged and in nay case the more "soluble" fraction, according to your data, seems to be less than 8% of total?

6.- If (as shown in fig 4), mtDNA damage by ROS promotes degradation, and repair is less effective in the mutants, they should show higher degradation and hence lower levels of mtDNA. The alternative is a higher rate of synthesis (in summary, higher turnover). Please comment on that and how it could be addressed in future works.

7.- Figure 7: there seems to be a higher SMAC level in mutants in basal conditions. Is that correct? Could the blots be quantified to show the changes in SMAC normalized by Hsp90?

8.- For the proper interpretation of suppl. figures 7 and 8 a quantification of the basal level of cyt c comparing controls and patients (by Western blot) and, if possible, a quantification of the cyt c release (obtained from the cell pictures) would be appreciated.

1st Revision - authors' response

02 October 2015

All the comments of the reviewers have been taken into account. Among all the requested experiments, some have been added in the revised version of the manuscript, others are presented in response to the reviewers in order to not overload the article that already contains many figures.

We also propose a novel title « *CHCHD10* mutations promote loss of mitochondrial cristae junctions with defect in mitochondrial genome maintenance and apoptosis » that seems to better reflect the content of the article.

Referee # 1

1. "In Figure 1E, the association of *CHCHD10* to *Mic60* is shown by co-immunoprecipitation. To exclude unspecific co-elution of *CHCHD10* with *Mic60* IP the authors have to include a control like preimmunserum. Furthermore, it should be given in

the figure legend how many percentage is loaded for the input and elution fractions to allow assessment of the efficiency of the pulldown."

We do agree with reviewer 1 (and reviewer 3) that it is important to further validate the co-immunoprecipitation experiments. A negative control with IgG was added in the co-immunoprecipitation experiments performed with an antibody against Mic60/Mitofilin. In control and patient fibroblasts, Mic60/Mitofilin was co-immunoprecipitated with CHCHD10 and also with Mic19/CHCHD3, another component of the MICOS complex. Figure 1E has been replaced by a new one (Fig.2A). The amounts of proteins are indicated in the figure legend.

P.5, L.15: We then evaluated whether CHCHD10 could physically interact with mitofilin. We found that CHCHD10 and CHCHD3 were immunoprecipitated by a rabbit polyclonal anti-mitofilin antibody both in control and patient fibroblasts (Fig.2A).

P.25, L.11: Figure 2. Interaction of CHCHD10 with components of MICOS complex. A. Co-immunoprecipitation (IP) of endogenous mitofilin, CHCHD10 and CHCHD3 in control and patient fibroblasts (1mg of total extracts were used for each IP (lower panel) and 200µg were used for the input (upper panel)). The same results were found in P2 fibroblasts (not shown).

2. *"To substantiate the association of CHCHD10 to Mic60 the authors should perform reverse pulldown experiments, e.g. using Flag-tagged CHCHD10 of HeLa cells for co-IPs."*

We performed reverse co-immunoprecipitation experiments with an antibody against CHCHD10 in HeLa cells and found that CHCHD10 was co-immunoprecipitated with Mic60/Mitofilin. We thought that it would be more convincing to use endogenous proteins rather than overexpressed Flag-tagged CHCHD10. To maintain consistency in terms of the cell types used, we did not add these results in Figure 2 but they are presented below for the benefit of the reviewers.

Co-immunoprecipitation of endogenous CHCHD10 and Mitofilin in HeLa cells (right panel). 1mg of total extracts were used for each immunoprecipitation and 50µg were used for the input (left panel).

3. *"The authors present a small section of a 2D PAGE to reveal co-migration of MICOS components and CHCHD10 (Figure 1A, C). To judge whether the co-migration is specific the whole second dimension has to be shown for each protein. Furthermore, the size of the markers of the first dimension should be indicated. Is it possible to shift the Mic60 signal on BN/2D in Flag-CHCHD10 using anti-Flag antibodies? Such or similar experiments would demonstrate the presence of Mic60 in the same complex as CHCHD10."*

The whole second dimensions are presented below :

A. BN-PAGE of mouse brain mitochondria. 300 μ g of isolated mitochondria from mouse brain were solubilized either with digitonin (6g/g) or with β -D-lauryl maltoside (1.8%). **B.** 2D BN-PAGE. Membranes are shown in the order of incubated antibodies.

2D BN-PAGE from control (left panel) and P1 (right panel) fibroblasts solubilized with two rounds of digitonin (8g/g and 6g/g w/v). The protein levels of assembled CHCHD10 and CHCHD3 were decreased in patient fibroblasts and accumulation of intermediates of assembly (I.A.) could be observed in mutant CHCHD10 fibroblasts.

We did not observe any appreciable differences in the Mic60 migration when we overexpressed wt and mutant Flag-CHCHD10 plasmids. Unfortunately, we got a very faint signal with Flag-Ab on BN/2D unlike what is presented below.

Western blot (left panel) and 1D/BN-PAGE (right panel) showing MICOS (anti-Mitofilin Ab) and OXPHOS complexes in untransfected (mock) and transfected cells with wild-type (wt) or S59L constructs.

4. "In Figure 1C the migration pattern of Mic60 remains unaltered in patients mitochondria. Does this mean that CHCHD10 is not important for structural integrity of MICOS? Can the authors speculate about the role of CHCHD10 within the MICOS complex?"

Ott and colleagues recently suggested a hierarchy of MICOS subunits (Ott et al., 2015). Mic60/Mitofilin and Mic19/CHCD3 could be key components of the MICOS complex since their depletion dramatically affects other MICOS components and the morphology of mitochondrial cristae. The other MICOS components that were tested, Mic25/CHCHD6, Mic27/ApoO and Mic23/ApoO appear to be periphery components because their deletion does not affect the remaining MICOS component nor cristae morphology. With the solubilization conditions that were used in our experiments, the CHCHD10 mutant allele does not seem to alter Mic60 migration pattern. Our results therefore suggest that CHCHD10 could be a peripheral protein. This is further corroborated by the observation that CHCHD10 is soluble in the IMS. However, both HeLa cells overexpressing Ser59Leu mutant and patient fibroblasts have abnormal mitochondrial cristae with loss of cristae junctions. At this stage, we cannot be definitive about the importance of CHCHD10 in maintaining the structural integrity of the MICOS complex. Nevertheless, our results are consistent with CHCHD10 mutations having a detrimental impact on MICOS function.

5. "In Figure 1B the complete 1st dimension should be shown for each blot".

The complete first dimension of gels is shown with sequential labelling in the following order: Mitofilin, CI, CIV, CIII and CII.

6. "The labeling of Figure 1D is not entirely clear. Which bands correspond to the labeled respiratory chain complexes?"

The labeling of the new Figure 1D has been clarified. Because of the size of the image, the figure 1 was separated into two.

7. "The authors should show steady state protein levels of mitochondrial-encoded proteins and respiratory chain components by SDS-PAGE. This analysis will reveal whether the reduced copy number of mitochondrial nucleoids affects levels of mitochondrial-encoded proteins."

We agree with the reviewer that it is important to address this question. The blot, showing that levels of proteins encoded by mtDNA are not affected, is shown in supplementary figure 5. An additional sentence has been added in the manuscript to more clearly provide this information to readers.

P.6, L.15: Thus, the reduction of nucleoid number is not related to a reduction in the amount of mtDNA. Furthermore, it does not lead to a decrease in expression level of proteins encoded by mtDNA in patient fibroblasts (supplementary Fig.5).

P.37, L.16: Supplementary figure 5. Expression level of OXPHOS proteins, encoded either by mtDNA or by nuclear genes, is not affected in patient fibroblasts. Representative western blot of ATP5A, UQCRC2, COX I, SDHB, COX II and NDUFB8 proteins performed with fibroblast lysates obtained from control (C) and patients (P1, P2). Hsp60 and GAPDH are also shown as controls. ns, non specific; * indicates proteins encoded by mtDNA.

8. "Does CHCHD10 show any sequence similarity to known MICOS components? Since MICOS subunits are conserved from yeast to human it would be interesting to see whether CHCHD10 is conserved as well".

MICOS subunits are conserved from yeast to human but there is no CHCHD10 ortholog in yeast. CHCHD10 is conserved in chimpanzee, dog, cow, mouse, rat, chicken, zebrafish, and frog. In humans, there is no significant similarity between CHCHD10 and other MICOS components. However, this lack of conservation also applies to other confirmed MICOS components.

Referee #2

1. "Fig 1C in which CHCHD10 is described as reduced in the P1 and P2 samples is not of the best quality. Each sample is made up of 4 separate sections, which for P1 and 2 overlap the CHCHD6 signal partially obscuring it. Because the panels for each antibody are split into separate sample panels, it is not possible to tell if all were exposed to the detection system for the same length of time or if the difference in signal is a result of different exposure times rather than reflecting the protein amounts. This is particularly important when claiming that the levels of CHCHD10 are down whilst those of CHCHD6 are up. The current data is not suitable to make the conclusions presented".

A full version of Fig.1C has now been provided to answer to the comments from the first reviewer. Please, see answer to comment n°3 from referee #1.

The blot for the mouse seemed to be able to detect all 4 antibody signals at the same time. A similar panel with all control and patient samples detected together would be needed to make this data convincing.

This is a misinterpretation because the detection was sequential. We hope that our clarifications to comment n°3 from Referee #1 have clarified the query raised by the reviewer.

2. "The results for Patient 1 and 2 differ. This is not altogether surprising but they must be consistent if the message is that they both differ in a similar way to controls. Fig 2 depicts quantification indicating that there is a significant difference between both patients and control for nucleoid number but only a significant difference in size of nucleoids for Patient 1."

We agree that there are slight differences when comparing the data obtained with fibroblasts from the 2 patients. Despite being siblings carrying the same mutation, there were differences in clinical presentations and disease severity. Fibroblasts were obtained after 10 years of disease onset for Patient 1 and after 2 years of disease onset for Patient 2. Notwithstanding these differences, both cell lines showed the same consistent abnormalities, namely :

- respiratory chain deficiency
- abnormal mitochondria by EM
- fragmentation of the mitochondrial network with no fusion defect
- disassembled MICOS complex
- inhibition of apoptosis

With regards to mtDNA, both cell lines showed a decrease of nucleoid number compared with control fibroblasts. We agree that we observed a slight increase of nucleoid surface in cells from Patient 1 with anti-DNA antibodies (New Fig.3C). However, analysis of patient cells with anti-TFAM antibodies (sup Fig.4) and overexpression of mutant alleles in HeLa cells (Fig.4) confirmed the absence of increased nucleoid surface and of nucleoid aggregation.

3. "Fig 4 gives tabulated values for the mtDNA in the soluble vs pellet fraction. This is a meaningless, without as a minimum, a western to show that the fractionation is consistent between samples. It does separate the nuclear from the cytoplasmic component. This will not discriminate between cytosolic and mitochondrial compartments. It may be more of an indication of fractured mitochondria rather than disturbed nucleoids".

The Western blots have been done but they were not shown in our original submission. The novel figure 5A shows mitochondrial fractions isolated from control and patient fibroblasts and the sub-fractionating into supernatant and pellet. The hybridization of both blots with specific antibodies shows the absence of contamination of the different compartments. The corresponding legend has been added.

P.26, L.20: Figure 5. Nucleoid disorganization in patient fibroblasts leading to a defect in mtDNA repair under conditions of oxidative stress. A. Total extracts from control (C) fibroblasts and intact isolated mitochondria from control and patient fibroblasts

(P1, P2) were analyzed by immunoblotting using antibodies against PCNA (nuclear protein), GAPDH (cytosolic protein) or SMAC (mitochondrial intermembrane space protein) (upper panel). Mitochondria from patient and control fibroblasts were incubated with NP-40 and separated into pellets (P) and supernatants (S). The fractions of each extraction were subjected to western blot analysis. VDAC and SMAC were used to identify behaviors of well defined mitochondrial proteins that are integral membrane and soluble proteins, respectively (middle panel). Ratio of mtDNA amplified from supernatant/mtDNA amplified from pellet, by qPCR, was quantified in control and patient fibroblasts (lower panel).

4. *"Levels of LC3 are shown graphically and not the westerns. It is necessary to see the primary data to evaluate the validity of the graphs".*

The Western blots have been provided in figure 5A. Figure 5B shows the quantitative analysis of the cells under basal conditions without chloroquine (-CLQ) and figure 5C shows the quantitative analysis of LC3 levels after chloroquine treatment (+CLQ). Please note that figure 5 corresponds to figure 6 in the revised version.

5. *"Controls have a level of natural variation as can be seen in fig 6D where the difference in O/N STS is high. The trend following treatments is identical between patients and controls, only the absolute % of depolarized cells varies".*

We fully agree with the reviewer that there is some (expected) natural variability when using control cells and for this reason we chose to use 2 independent control cell lines. The two patient cell lines are following the same trend and they do demonstrate significantly less depolarized mitochondria following STS treatment. This is consistent with the decrease in caspase dependent activity observed in those cells. Taken together, our results indicate that the *CHCHD10* patient cells are less sensitive than control cells to apoptosis.

Minor comments

6. *"In the introduction there is a rather awkward sentence that is not readily understandable" If the involvement of nuclear genes responsible for either mtDNA replication/repair or nucleotide pool maintenance was predictable, that of genes falling in other categories such as MFN2, which is involved in mitochondrial fusion, was rather unexpected."*

Genes encoding for proteins involved in mtDNA metabolism or in the maintenance of the intra mitochondrial nucleotide pool were obvious candidates for mtDNA instability disorders. On the contrary, the link between defect in mitochondrial dynamics and mtDNA instability was not suspected until recently. Taking on board the comments of the reviewer, this specific sentence has been clarified.

P.3, L.5: It was predictable that nuclear genes involved in mtDNA replication and repair, or the maintenance of the intramitochondrial nucleotide pool would be implicated in human disorders characterised by mtDNA instability. However, the link between disturbed mitochondrial dynamics and human disease only became apparent with the description of the varied neurological manifestations associated with *MFN2* and *OPA1* mutations.

7. *"Fig 1 B has a horizontal arrow for kDa. This is not necessary or accurate as this represents a 1D BN PAGE expt."*

We fully agree with the reviewer and the figure has been modified accordingly.

8. *"Quality of the CHCHD10 panel in Fig1E - pre clear is poor, especially compared to the adjacent panel of the IP. The indication of data not shown for P2 should be in the main text, not the legend".*

Additional co-immunoprecipitation experiments have been performed and the figure 1E has been replaced by figure 2A. Please refer to our reply to comment n°1 from referee #1.

9. "Fig 2 A - this may be a well known phenomenon but if the antibody is against DNA why does the nucleus not light up ?"

The possible explanations are the fixation procedure (we used 4% PFA instead of TCA or methanol which are recommended) or to the antigen retrieval method used. It is also possible that the antigen is masked by chromatin or nuclear proteins. Furthermore, the antibody used is highly sensitive for the detection of mitochondrial DNA and the signal emitted from mtDNA is probably much stronger because mtDNA is not packed as densely as nuclear DNA. Under the same conditions of fixation with the same antibody, no nuclear signal has been observed in several articles (for example, see Di Re *et al*, Nucl Acids Research, 2009).

10. "FLAG should be given in capital letters."

This has been corrected both in the text and in the figures.

11. "I don't really understand the axis label for the mtDNA content. "Data were expressed as ratio between mtDNA and nDNA concentration." What is regarded as a control normal value ? Do they mean concentration or amount ? In Fig2 the control is set as 1, but in Fig 3 none of the samples are given a value of 1, so what are they compared to ?"

The determination of mtDNA copy number in cells was performed by qPCR using TaqMan probes specific for mtDNA (12S rRNA gene) and nuclear DNA (*MLH1* gene) (Bannwarth *et al.*, 2012). The ratio mtDNA amount/nuclear DNA amount was used as a measure of mtDNA copy number that was compared in control and patient cells (new Fig. 3 and 4).

Referee #3:

General considerations:

1. "I assume that the fibroblast and HeLa cell results are normally from glucose medium culture. The fibroblast model grown in glucose could be not completely appropriate to analyze some phenotypes associated with the mutations in *CHCHD10*, that seem to be cell-type specific. For example the deletions and complex V alterations are seen in patient muscle but not in fibroblasts. By the same token, the respiratory defect is not evidenced in fibroblasts grown in glucose and the differences in DEVDase activity after apoptosis induction, very clear in fibroblasts, are not detected in HeLa cells. Since the respiratory defect, although relatively mild, could promote a metabolic shift to a more glycolytic profile in the mutant cells, it would be interesting to see some of the analysis performed in the galactose growth medium (for example the membrane potential analysis). In this respect, do the average cell size or cell morphology change when comparing control and patient fibroblasts? And between the glucose and galactose medium?"

In a glucose-free medium containing galactose, cells are forced to rely predominantly on OXPHOS for ATP production because galactose feeds the glycolytic pathway with a low efficiency. We fully agree that galactose medium could unmask a defect that is compensated in glucose medium. We therefore performed novel experiments in HeLa cells overexpressing *CHCHD10* mutant alleles to compare (i) DEVD-ase activity, (ii) Annexin V/DAPI staining and, (iii) cell morphology in glucose and galactose medium. Analysis of cell morphology was also compared in control and patient fibroblasts and between the glucose and galactose medium.

Transfections in HeLa cells were performed with empty vector or vectors encoding either wild-type CHCHD10 or mutant CHCHD10 (P34S or S59L). HeLa cells in glucose or in galactose medium were treated with 1 μM Actinomycin D for 2, 4 or 8h with measurement of DEVD-ase activity.

Transfections in HeLa cells were performed with empty vector or vectors encoding either wild-type CHCHD10 or mutant CHCHD10 (P34S or S59L). HeLa cells in glucose (left panel) or in galactose (right panel) medium were treated with 1 μM Actinomycin D for 2, 4 or 8h with measurement of Annexin V/DAPI staining.

In conclusion, we obtained similar results as in Figure 7 in glucose medium. In our hands, transfected HeLa cells do not survive in galactose medium and die through necrosis (with no caspase activation and DAPI positive staining). Under these culture conditions, we cannot conclude about cell sensitivity to actinomycin D.

Flow cytometry analysis of control (C1, C2) and patient (P1, P2) fibroblasts (A) and of HeLa cells expressing the empty vector or vectors encoding either wild-type CHCHD10 or mutant CHCHD10 (P34S or S59L) (B) in glucose or in galactose medium. Cell size and granularity of fibroblasts and HeLa cells are not significantly modified by the expression of mutant alleles both in glucose and galactose medium.

2. "One of the conclusions of this work is that mitochondrial fusion is not affected by the mutations in CHCHD10. This is somehow unexpected in the view of the role proposed for

this protein and the fact, for example, that the KO for a CHCHD10 partner, CHCHD3 (also reduced in the mutants) does present fusion alterations (Darshi et al, JBC, 2011). Since diffusion of components of the inner membrane as well as those of the nucleoids is slow (Busch et al, Phil. Trans. R. Soc. B 369: 20130442. <http://dx.doi.org/10.1098/rstb.2013.0442>, 2014), may be longer times are required to detect differences in the fusion process. In this respect, P2 seems to have at longer times a reduction in fusion (suppl. figure 2). Please comment on this. And, how do you explain the mt fragmentation pattern in the mutants, by an increase in fission? Some more work could be done in future works to clarify this point."

We are grateful for the Reviewer's insightful comments. To rigorously address these points, we have conducted additional work to assess mitochondrial morphology after blocking mitochondrial division in the revised manuscript (Supplementary Fig. 3), as performed by Darschi et al. (JBC, 286: 2918-32. 2011). We ectopically expressed a dominant negative form of Drp1 (Drp1^{K38A}) and elongated mitochondria were found in both control and patient cells. These additional results confirm that mitochondrial fusion is not blocked in CHCHD10 patient cells. It appears that although CHCHD10 and CHCHD3 interact, these proteins play different roles in mitochondrial fusion. In terms of the fusion assay in supplementary Fig.2, the apparent difference between the control and patient 2 cells is not significant. Therefore, we think that mitochondrial fusion is not decreased in these patient cells. In addition, we fully agree with the Reviewer that mitochondrial fragmentation in the patient cells could result from increased mitochondrial division. This hypothesis will need to be addressed as part of future studies.

P.5, L.4: To confirm the absence of fusion defect in patient fibroblasts, we ectopically expressed a dominant negative mutant of the fission protein Drp1 (Drp1^{K38A}) (Smirnova et al, J Cell Biol 1998), and found that the control and patient cells similarly elongate mitochondria (supplementary Fig.3).

P.37, L.1: Supplementary figure 3. CHCHD10 mutant fibroblasts are not defective in mitochondrial fusion. Control and patient cells were infected with lentiviruses expressing wild-type Drp1 or a dominant negative Drp1^{K38A}. Cells were subjected to immunofluorescence microscopy using antibodies to Drp1 (red) and the mitochondrial protein Tom20 (green). The expression of Drp1^{K38A} blocked mitochondrial division and elongated mitochondrial tubules in the control and patient cells. Nuclei were visualized by DAPI staining (blue). Boxed regions show magnified images. Scale bar: 20 µm.

3. *"Another important conclusion in the manuscript is that CHCHD10 is a component of the MICOS complex. In a recent article, Guarani et al (eLife 2015;4:e06265. DOI: 10.7554/eLife.06265) using interaction proteomics, systematically screened for proteins that associate with cristae junctions and although they found most known and some new interactions, failed to detect CHCHD10. Although the results presented are consistent with the proposal, this could be reinforced (see also specific points). For example, the co-immunoprecipitation experiment (Fig 1E) could be done also with the anti-CHCHD10 antibody to see if mitofilin or other MICOS components are pulled-down".*

It is true that CHCHD10 was not detected by Guarani and colleagues in their systematic proteomic analysis of the MICOS complex. However, they also failed to detect the transmembrane protein Mic10, another known MICOS component, possibly due to its small size. Mic10 is 4KDa smaller than CHCHD10. We do agree that it was important to reinforce our data and we also performed reverse co-immunoprecipitation experiments in HeLa cells with an antibody against CHCHD10 and found that Mitofilin was pulled-down (please, see answer to comment n°2 from referee #1).

Specific points:

1. *"Fig 1 A. It is interesting to see tissue data in addition to cell culture, but It would be interesting to know why have they chosen mouse brain for the 2D and what is the size of the MICOS complex band containing mitofilin and CHCHD10".*

Patients carrying *CHCHD10* mutations present with frontotemporal dementia and unfortunately, we do not have access to post mortem brain tissue from affected individuals for further analysis. For this specific reason, it was interesting to look at mouse brain. The size (700 KDa) was added on the figure 1A.

2. *"Fig 1B. In the case of P2 sample, I have the impression that if normalization is made by the CII level, no big decrease in MICOS would be found. Is that correct?"*

We agree with the reviewer that MICOS is more stable steadily in Patient 2 than in Patient 1. Fig.1B also suggests that CIV assembly defect is lower in Patient 2. It is not uncommon to observe phenotypic differences between patients carrying the same pathogenic mutation, presumably due to the influence of modifier genes and environmental factors. This is the case for Patients 1 and 2, but as detailed in our reply to comment n°2 from referee #2, both patient cell lines differ in a similar way to controls.

3. *"Fig 1C. The proportion of CHCHD3 vs D6 changes as well as the co-localization of bands. Thus, co-migration seems total in C but only partial in P1 (more clear) and P2. How are these changes in the proportion of the different components compatible with the MICOS complex keeping the same size?"*

To better clarify this point, we have run C and P1 strips of the first native dimension in a large denaturing 2D-gel and we clearly observed that mutant P1 has an accumulation of intermediate species of assembly after immunoblotting for CHCHD6 and CHCHD10. In relation to the size in the first dimension, we hypothesize that both CHCHD6 and CHCHD10 are peripheral proteins within MICOS. The loss of ~25kDa may not be detectable compared with the approximate size of the fully assembled complex of ~700kDa, using our electrophoresis conditions. It should also be kept in mind that part of the complex remains fully assembled since there is still production of the WT protein.

4. *"Fig 1 D. In the analysis of supercomplexes there is a low resolution (separation) of high MW bands (may be this is related to the electrophoresis conditions, that are very different to the original ones proposed by Schagger). The more clear effect in complex levels is the strong reduction in free CIV (also evident for patient 1 in fig 1B). I am surprised that this reduction has no clear correlation with the complex activities shown in suppl. Table 1. In particular, I find unexpected that P1 (Bannwarth et al. 2014) shows a higher activity in COX than P2 having a more severe reduction in CIV. It would be interesting to see also here data on coupled/uncoupled respiration and in galactose medium".*

In Fig.1D, the cell lines from both patients were found to have a CIV assembly defect that is also found in Fig.1B with Patient 1 having an apparently greater defect compared with Patient 2. Both patients also have a complex IV deficiency by spectrophotometry under galactose medium with 167.1 and 146.8 nmol/min/mg of proteins for P1 and P2, respectively (181.7 < control values <315.4). These 2 values (167.1 versus 146.8) are comparable and it is difficult to conclude that there is a significant difference. Furthermore, in both cases, COX activity was normal under glucose medium conditions suggesting that there is no major biochemical defect. Taking on board the Reviewer's comments, we performed polarographic analysis for Patient 2 and no abnormality in glucose medium was found as for Patient 1 (New Sup.Table 1). In galactose medium, we found no decrease of oxygen consumption for P2 and this result could be consistent with a lower assembly defect than the one presented by P1. However, in our experience, there is frequently no clear correlation between assembly and activity of RC complexes in patients with mitochondrial disorders. As mentioned by Reviewer#1, it is not altogether surprising that the 2 patients differ and the best conclusion that we can draw is that both patients present a defect in COX assembly with a CIV deficiency by spectrophotometry in galactose medium only.

5. *"Figure 2-3, (analysis of nucleoids): Why anti-DNA antibody detect less dots in mutant cells if total mtDNA is unchanged and in any case the more "soluble" fraction, according to your data, seems to be less than 8% of total?"*

Anti-DNA and anti-TFAM antibodies detected a decrease of nucleoid number both in patient fibroblasts and in HeLa cells expressing *CHCHD10* mutant alleles without a reduction in mtDNA copy number. The new figure 5A, that has been provided, shows that mtDNA is found in the soluble fraction of patient fibroblasts (8 and 4 times more than for control in P1 and P2 respectively) and that the inverse correlation between the nucleoid number and the amount of mtDNA in the soluble fraction varies in a similar way for the 2 patients. We agree with the reviewer that we would expect more "soluble" mtDNA depending on the number of nucleoid dots in mutant cells. However, dots recorded by confocal analysis correspond to clusters of structures that are probably disassembled by disorganization of cristae. This disorganization is likely to lead to the separation of the clusters with a release of mtDNA in the matrix and also with mtDNA copies always associated to the membrane but not detected by confocal analysis due to cluster loss.

6. *"If (as shown in fig 4), mtDNA damage by ROS promotes degradation, and repair is less effective in the mutants, they should show higher degradation and hence lower levels of mtDNA. The alternative is a higher rate of synthesis (in summary, higher turnover). Please comment on that and how it could be addressed in future works."*

In muscle of patients with mitochondrial disease, the respiratory chain defect is frequently compensated by proliferation of subsarcolemmal mitochondria. As part of future work, it will be very interesting to determine whether mitochondrial mass is increased in patients' fibroblasts compared with control cells, both under basal conditions and after oxidative stress.

7. *"Figure 7: there seems to be a higher SMAC level in mutants in basal conditions. Is that correct? Could the blots be quantified to show the changes in SMAC normalized by Hsp90?"*

We do agree with the reviewer that, under basal condition, SMAC level seems higher in cells overexpressing mutants on the blot presented in Figure 7. Another blot is presented below and the quantitative analysis indicate that there is no significant difference.

Upper panel. Mitochondrial outer membrane permeabilization in HeLa cells overexpressing *CHCHD10* mutant alleles was determined by western blot by assessing the degradation of Smac. Hsp90 was used as a loading control. Lower panel. Quantitative analysis of Smac/Hsp90 ratio from immunoblots (n=2).

8. "For the proper interpretation of suppl. figures 7 and 8, a quantification of the basal level of cyt c comparing controls and patients (by Western blot) and, if possible, a quantification of the cyt c release (obtained from the cell pictures) would be appreciated."

We have performed a quantitative analysis of the basal expression of cyt c by Western blot in : (i) control and patient fibroblasts, and (ii) in HeLa cells overexpressing the empty vector (EV), wild-type (WT) *CHCHD10* and the 2 mutant forms (P34S and S59L). We found no significant differences in both cases and the additional data have been presented below. We did not quantify the cyt c release from the cell that were pictured because we cannot reproducibly measure an overall fluorescence without internal control.

Control and patient fibroblasts

Left panel. Immunoblot of *cyt c* expression in fibroblasts of controls (C1, C2) and patients (P1, P2). Right panel. Quantitative analysis of *cyt c*/Hsp90 ratio from immunoblots ($n=3$). ns, not significant.

HeLa cells

Left panel. Immunoblot of *cyt c* expression in HeLa cells overexpressing the empty vector (EV), wild-type (WT) CHCHD10, P34S or S59L mutant alleles. Right panel. Quantitative analysis of *cyt c*/Hsp90 ratio from immunoblots ($n=2$). ns, not significant.

Thank you for the submission of your revised manuscript to EMBO Molecular Medicine. We have now received the enclosed reports from the Reviewers that were asked to re-assess it.

As you will see while Reviewers are globally supportive, all three have a few remaining issues that, quite clearly, reflect issues with presentation, completeness and quality of figures and doubts in interpretation.

We very much agree with the reviewers' points (that should be fully and satisfactorily addressed). Indeed I would like to express a number of additional editorial concerns, some of which partially overlap with those from the reviewers.

1) There are multiple horizontal and vertical splices (ex-post juxtaposition of portions of gels/blots originally separated in location) in many of the figures. Some of these are so obvious that it is clear that these are intended to be evident to the reader. Others are instead subtler and only revealed after our pre-publishing quality control and image screening routines. For both scenarios, the splices must be made clear with a white space or black separation line on the figures and a mention in the figure legends. In addition, please provide us with an explanation of these occurrences and the full source data set for the figures with accurate mapping of the source data elements to the submitted final figures so that we can cross-check.

2) There are missing scale bars in many figures, including the supplemental ones. Please add them.

3) Many figures are of less than optimal quality, and appear blocky/blurry when magnifying (Figs 2, 5, 8 and others). It would be better if you could use higher resolution images in your figures.

Please also carefully consider the following final general Editorial amendments/requests to be include in your revision:

1) As per our Author Guidelines, the description of all reported data that includes statistical testing must state the name of the statistical test used to generate error bars and P values, the number (n) of independent experiments underlying each data point (not replicate measures of one sample), and the actual P value for each test (not merely 'significant' or ' $P < 0.05$ '). I note that you have not provided the exact P values in all cases.

2) Please incorporate the "Funding" section into the "Acknowledgments" one.

3) For experiments involving human subjects the authors must identify the committee approving the experiments and include a statement that informed consent was obtained from all subjects and that the experiments conformed to the principles set out in the WMA Declaration of Helsinki [<http://www.wma.net/en/30publications/10policies/b3/>] and the NIH Belmont Report [<http://ohsr.od.nih.gov/guidelines/belmont.html>]. Any restrictions on the availability or on the use of human data or samples should be clearly specified in the manuscript. Any restrictions that may detract from the overall impact of a study or undermine its reproducibility will be taken into account in the editorial decision. I note that there is no specific mention of the source and culture conditions etc of the human cells used in your experiments (to simply refer to previous work is not sufficient in this case. I also note that although you mentioned the use of skin biopsies in the author checklist, section C, you did not provide any information in section E.

4) The supplemental information text is currently included in the manuscript file. Please remove and combine with the figures in a separate file, strictly according to our current guidelines for supplementary information (Expanded View: <http://embomolmed.embopress.org/authorguide#expandedview>). Please also amend callouts in the manuscript accordingly.

5) Every published paper now includes a 'Synopsis' to further enhance discoverability. Synopses are displayed on the journal webpage and are freely accessible to all readers. They include a short standfirst (to be written by the editor) as well as 2-5 one sentence bullet points that summarise the paper (to be written by the author). Please provide the short list of bullet points that summarise the

key NEW findings. The bullet points should be designed to be complementary to the abstract - i.e. not repeat the same text. We encourage inclusion of key acronyms and quantitative information. Please use the passive voice. Please attach these in a separate file or send them by email, we will incorporate them accordingly.

Additional information for submission is provided below.

I look forward to receiving the next, final revised form of your manuscript as soon as possible

***** Reviewer's comments *****

Referee #1 (Remarks):

The revised version of the manuscript by Genin and colleagues satisfactorily addressed my concerns. The findings are interesting and well presented. I recommend some minor modification of the manuscript before publication of the revised version in EMBO Mol Med.:

In the novel Fig. 2A the authors should use the same exposition of the gels for input and elution fractions (for instance for CHCHD10). Otherwise, it is hard to assess the efficiency of the pulldown experiment.

To support the association between Mitofilin and CHCHD10 the authors included a reverse co-immunoprecipitation with anti-CHCHD10 antibodies in the response letter. I strongly recommend presenting this blot in Fig. 2. The data set is important to strengthen the author's conclusion that Mitofilin interact with CHCHD10.

Referee #2 (Remarks):

The authors have clearly spent some time addressing the concerns of the reviewers and some of these problems have been resolved. There are still some outstanding issues that have not been fully clarified. The selection of data presented in the rebuttal and the manuscript do not always seem in agreement.

Referee #3 (Remarks):

The manuscript by Genin et al., has improved with the new experiments and controls included and most of the concerns raised by the reviewers have been, in my opinion, adequately answered.

I have only a couple of minor remarks :

1) In figure 1 D, complex I is marked in a position that doesn't correspond with its molecular size. CI migrates, in BN gels, above complex V (usually visible in the gels even without further staining) and above SC III2-IV. In addition, complex III, even in DDM gels, corresponds to a dimer (panel B in figure 1 could be marked accordingly).

Below are the sizes (kDa) expected for complexes in bovine heart mitochondria:

Monomeric C I: 1000
 Monomeric complex V: 597
 Dimeric C III: 482
 Monomeric C IV: 205
 Monomeric C II: 123

I ignore what is the nature of the band marked as CI in figure 1D but, again, fully assembled CI

should be much closer to the SCs, slightly above SC III2-IV. Please, compare with the position of CI in the panel A of the figure presented on page 2 of the rebuttal letter and its relative migration with respect to MICOS complex (here CI is well marked for digitonin solubilization) or on page 4 of the same letter (DDM solubilization).

2) Migration of CHCHD6 compared to the markers varies between the blots presented in the rebuttal letter and in figure 1.

2nd Revision - authors' response

29 October 2015

All the comments of the reviewers have been taken into account.

Referee #1

- 1. « *In the novel Fig. 2A the authors should use the same exposition of the gels for input and elution fractions (for instance for CHCHD10). Otherwise, it is hard to assess the efficiency of the pulldown experiment* ».

We agree and this has been done in the Fig.2A

- 2. « *To support the association between Mitofilin and CHCHD10 the authors included a reverse co-immunoprecipitation with anti-CHCHD10 antibodies in the response letter. I strongly recommend presenting this blot in Fig. 2. The data set is important to strengthen the author's conclusion that Mitofilin interact with CHCHD10* ».

The reverse co-immunoprecipitation experiment has been added in the novel version of Fig. 2

Referee #3

1. « *In figure 1 D, complex I is marked in a position that doesn't correspond with its molecular size. CI migrates, in BN gels, above complex V (usually visible in the gels even without further staining) and above SC III2-IV. In addition, complex III, even in DDM gels, corresponds to a dimer (panel B in figure 1 could be marked accordingly).*

Below are the sizes (kDa) expected for complexes in bovine heart mitochondria:

Monomeric C I: 1000

Monomeric complex V: 597

Dimeric C III: 482

Monomeric C IV: 205

Monomeric C II: 123

I ignore what is the nature of the band marked as CI in figure 1D but, again, fully assembled CI should be much closer to the SCs, slightly above SC III2-IV. Please, compare with the position of CI in the panel A of the figure presented on page 2 of the rebuttal letter and its relative migration with respect to MICOS complex (here CI is well marked for digitonin solubilization) or on page 4 of the same letter (DDM solubilization) ».

We fully agree with the reviewer and there was a mistake in Fig.1D. The blots were labeled with two different antibodies against CI (NDUFS3 and NDUFS9). The upper band, corresponding to CI, is found in both cases. The band, annotated CI by mistake, corresponds to an unspecific signal. In the novel version of the Fig.1D, the « NDUFS9 » blot has been deleted and the position of CI on the « NDUFS3 » blot has been corrected.

2. « Migration of CHCHD6 compared to the markers varies between the blots presented in the rebuttal letter and in figure 1 ».

In the first type of 2D-Blots we made 5-16% Bis-Tris gradient gels manually with a gradient mixer. To answer to reviewer's comments, we have used larger gels for 2D-WB. Since we don't have a large gradient mixer for these gels, we made a separating gel of 16% followed by a layer of 2 cm at 10% and another layer containing the strip of 10% non-denaturing gel. This is the case of the gels presented in the rebuttal letter. The differences in gel composition could explain the observed differences in protein migration.

1) There are multiple horizontal and vertical splices (ex-post juxtaposition of portions of gels/blots originally separated in location) in many of the figures. Some of these are so obvious that it is clear that these are intended to be evident to the reader. Others are instead subtler and only revealed after our pre-publishing quality control and image screening routines. For both scenarios, the splices must be made clear with a white space or black separation line on the figures and a mention in the figure legends. In addition, please provide us with an explanation of these occurrences and the full source data set for the figures with accurate mapping of the source data elements to the submitted final figures so that we can cross-check.

We agree that the dividing lines were missing in Fig. 1A and 1C. This has been corrected in the new figures and the raw data are presented below [Editor's note: OMITTED. The source data will be made fully available online upon publication].

In the panel of Fig. 2A, which corresponds to the IP, the negative control corresponding to the IP performed with IgG was located on the right of the original blot that is presented below Editor's note: OMITTED. The source data will be made fully available online upon publication]. For better consistency with the gel corresponding to the input, the IgG line has been deleted and put on the left of the IP blot. A dividing line has been drawn on the novel version of Fig.2A to clarify this point.

2) There are missing scale bars in many figures, including the supplemental ones. Please add them.

This has been done. Missing scale bars have been added in the Supplementary Figure 3 and in legends of

- Fig.2 : P27, L2
- Sup. Fig.1 : P8, L20 (doc expanded view)
- Sup. Fig.2 : P9, L11 (doc expanded view)
- Sup. Fig.7 : P10, L24 (doc expanded view)

3) Many figures are of less than optimal quality, and appear blocky/blurry when magnifying (Fig.s 2, 5, 8 and others). It would be better if you could use higher resolution images in your figures.

All figures were recorded in a new format with a better recording quality (1200 dpi minimum). Fig.9 has been reworked to improve quality.

Please also carefully consider the following final general Editorial amendments/requests to be include in your revision:

1) As per our Author Guidelines, the description of all reported data that includes statistical testing must state the name of the statistical test used to generate error bars and P values, the number (n) of independent experiments underlying each data point (not replicate measures of one sample), and the actual P value for each test (not merely 'significant' or ' $P < 0.05$ '). I note that you have not provided the exact P values in all cases.

This has been completed. The name of the statistical tests that were used has been added in the legend of

- Fig. 3: P27, L15
- Fig. 4: P28, L4
- Fig. 6: P29, L13
- Sup. Fig.2: P9, L9 (doc expanded view)

The number (n) of independent experiments underlying each data point has been added in the legend of

- Fig. 7: P29, L18 and 24
- Fig. 7: P30, L3
- Fig. 8: P30, L13

All the P values are found in the figure legends of the manuscript

- Fig. 2B: P27, L2
- Fig. 3B and C: P27, L10-12
- Fig. 4B: P28, L1
- Fig. 5C: P29, L3-4
- Fig. 7A, C and D: P29, L20-21; P30, L1,5 and 6
- Fig. 8B: P30, L16-17

and in the doc expanded view

- Sup. Fig.1C: P9, L2
- Sup. Fig.4B: P10, L2
- Sup. Fig.6C: P10, L20
- Sup. Fig.8B and C: P11, L11-12

2) Please incorporate the "Funding" section into the "Acknowledgments" one.

This has been done.

3) For experiments involving human subjects the authors must identify the committee approving the experiments and include a statement that informed consent was obtained from all subjects and that the experiments conformed to the principles set out in the WMA Declaration of Helsinki [<http://www.wma.net/en/30publications/10policies/b3/>] and the NIH Belmont Report [<http://ohsr.od.nih.gov/guidelines/belmont.html>]. Any restrictions on the availability or on the use of human data or samples should be clearly specified in the manuscript. Any restrictions that may detract from the overall impact of a study or undermine its reproducibility will be taken into account in the editorial decision. I note that there is no specific mention of the source and culture conditions etc of the human cells used in your experiments (to simply refer to previous work is not sufficient in this case. I also note that although you mentioned the use of skin biopsies in the author checklist, section C, you didnot provide any information in section E.

Missing information has been provided in the section E of the checklist:

"Skin punches were obtained after informed consent. All steps of experiments conformed to the principles set out in the WMA Declaration of Helsinki."

We also mentioned in the supplementary material (doc expanded view) that skin biopsies were obtained after informed consent: P1, L5.

4) The supplemental information text is currently included in the manuscript file. Please remove and combine with the figures in a separate file, strictly according to our current guidelines for supplementary information (Expanded View:

<http://embomolmed.embopress.org/authorguide#expandedview>). Please also amend callouts in the manuscript accordingly.

This has been done.

5) Every published paper now includes a 'Synopsis' to further enhance discoverability. Synopses are displayed on the journal webpage and are freely accessible to all readers. They include a short standfirst (to be written by the editor) as well as 2-5 one sentence bullet points that summarise the paper (to be written by the author). Please provide the short list of bullet points that summarise the key NEW findings. The bullet points should be designed to be complementary to the abstract - i.e. not repeat the same text. We encourage inclusion of key acronyms and quantitative information. Please use the passive voice. Please attach these in a separate file or send them by email, we will incorporate them accordingly.

A file called Synopsis, including a diagram, has been provided.

Last, 2 web sites have been added at the end of the manuscript and the Helvetica police has been inserted in all the figures.

3rd Editorial Decision

04 November 2015

I am now ready to accept your manuscript for publication.

Before I do however I would need the following from you:

1) We encourage the publication of source data, particularly for electrophoretic gels and blots, with the aim of making primary data more accessible and transparent to the reader. This would be appropriate here, given the (now clarified) splicing in your figure. This would imply a PDF file per figure that contains the original, uncropped and unprocessed scans of all or at least the key gels used in the manuscript. The PDF files should be labeled with the appropriate figure/panel number, and should have molecular weight markers; further annotation may be useful but is not essential. The PDF files will be published online with the article as supplementary "Source Data" files. If you have any questions regarding this just contact me.

2) Please add (very short) statements in the appropriate legends to declare and explain the juxtaposition of non-originally adjacent sections in the blots (Fig.s 1 and 2).

3) Fig 2B still appears to be missing a scale bar. Please amend.

Please provide updated files as soon as possible so that I can proceed with immediate acceptance.